

# Auroral meridian scanning photometer calibration using Jupiter

Brian J. Jackel[1], Craig Unick[1], Fokke Creutzberg[2], Greg Baker[1], Eric Davis[1], Eric F. Donovan[1],
Martin Connors[3], Cody Wilson[1], Jarrett Little[1], M. Greffen[1], and Neil McGuffin[1]

[1]University of Calgary, Alberta, Canada
[2]Natural Resources Canada Geomagnetism Laboratory
[3]Athabasca University, Alberta, Canada

*Correspondence to:* Brian J. Jackel
brian.jackel@ucalgary.ca

**Abstract.** Observations of astronomical sources provides information that can significantly enhance the utility of auroral data for scientific studies. Jupiter is used for field cross-calibration of 4 multi-spectral auroral meridian scanning photometers during 2011-15 northern hemisphere winters. Seasonal average optical field-of-view and local orientation estimates are obtained with uncertainties of $0.01°$ and $0.1°$ respectively. Estimates of absolute photometric sensitivity are repeatable to roughly 5%

from one month to the next, while the relative response between different wavelength channels is stable to better than 1%. Astronomical field calibrations and darkroom calibration differences are on the order of 10%. Atmospheric variability is the primary source of uncertainty; this may be reduced with data from co-located instruments such as all-sky imagers.

## 1 Introduction

Interactions between the solar wind and the terrestrial magnetic field produce a complex and dynamic geospace environment.

Ionospheric phenomena such as the aurora are connected to magnetospheric processes by mass and energy transport along magnetic field lines. Consequently, auroral observations provide information that can be used for remote sensing of distant plasma structure and dynamics. A single ground-based instrument can only view a small part of the global system. Larger scales may be studied with a combination of instruments at different locations (eg. Figure 1 and Table 1), but merging multiple data sets requires accurate information about device characteristics such as timing, orientation, and absolute spectral sensitivity.

Comprehensive calibration requires specialized equipment and skilled personnel that are typically available only at centrally located research facilities. With sufficient resources it is possible, at least in principle, to determine all device parameters that are required to convert raw instrument data numbers to physically useful quantities. Practical limitations can result in random or systematic uncertainties which may impede quantitative scientific analysis. This is particularly relevant for large networks of nominally identical instruments, where ongoing calibration of each device may be extremely challenging.

Even assuming ideal calibration at a central facility, many auroral instruments must be operated at remote field sites. Transfer between these locations requires a sequence of packing, shipping, and re-assembly that is time-consuming, costly, and may unintentionally alter instrument response. Furthermore, intermittent calibration cannot distinguish between a gradual drift or sudden changes.



**Figure 1.** Canadian meridian scanning photometer site locations (details in Table 1). Fan shapes indicate $4°$ optical beam width for altitudes of 110 and 220 km at elevations of $10°$ above the horizon. Contours indicate magnetic dipole latitude (IGRF 2010).

**Table 1.** Canadian meridian scanning photometer site information. Geographic latitude, longitude, and altitude are in degrees North, degrees East, and metres above mean sea level (WGS-84). L-shell and magnetic declination obtained from the IGRF model.

|  | Geographic | | | L-shell | | Declination | |
|---|---|---|---|---|---|---|---|
|  | Lat | Lon | Alt | 1988 | 2013 | 1988 | 2013 |
| RANK | 62.82 | 267.89 | 32 | 11.20 | 10.64 | -7.1 | -7.7 |
| GILL | 56.35 | 265.29 | 99 | 6.04 | 5.83 | 2.6 | -0.5 |
| PINA | 50.20 | 263.96 | 262 | 3.95 | 3.84 | 5.5 | 2.3 |
| FSMI | 60.02 | 248.05 | 205 | 6.65 | 6.58 | 24.3 | 15.8 |
| ATHA | 54.70 | 246.70 | 533 | 4.50 | 4.45 | 21.1 | 15.3 |




Extra-terrestrial sources, such as planets or stars, can be used for calibration of spatially resolved optical or radio frequency data. Instrument orientation can be determined from objects whose positions are well known, while source intensity can be used to verify instrument sensitivity. Astronomical sources are often detectable in existing auroral data streams, allowing for ongoing monitoring of system response and the possibility of retrospective re-analysis of older data sets. Practical application may be restricted by instrumental limitations and complications including man-made interference, clouds, aurora and other geophysical processes.

The focus of this paper is on the field calibration of a network of four auroral photometers using Jupiter as a standard reference. Some key features of optical aurorae are provided in Section 1.1, §1.2 introduces key calibration concepts and results, essential astronomical topics are presented in §1.3, and atmospheric effects are briefly reviewed in §1.4. An overview of instrument details is given in §2, data analysis and results are in §3, discussion in §4, followed by a summary and conclusions in §5.

## 1.1 Optical Aurora

In regions of geospace where magnetic field lines can be traced to the Earth, some charged particles may travel down to altitudes where neutral densities are no longer negligible. Collisions with atmospheric atoms or molecules may transfer energy which can be re-emitted as photons. Spectral, spatial, and temporal features of the optical aurora contain information about geospace plasma properties, allowing for remote sensing of magnetospheric topology and dynamics.

Auroral spectra are dominated by several extremely bright lines and bands from atomic oxygen and molecular nitrogen, with many other less intense features ranging from extreme ultra-violet through to far infra-red. The intensity of auroral emission at different wavelengths depends on precipitation energy and atmospheric composition, as more energetic particles are able to penetrate to lower altitudes where constituents may be more or less abundant. Consequently, observations at multiple wavelengths can be combined to infer characteristics of the precipitating particles (*Rees and Luckey*, 1974; *Strickland et al.*, 1989). These multi-spectral measurements can be challenging due to the wide dynamic range between very bright "green-line" (1-100 kiloRayleigh) emissions and extremely faint "proton aurora" ($<$ 100 Rayleighs).

Optical aurora typically occur within "auroral ovals", roughly centered around each geomagnetic pole, extending hundreds of kilometres in latitude and thousands of kilometres in longitude (*Akasofu*, 1965). Luminosity can be highly dynamic over a wide range of spatial scales, but quiet-time structures generally exhibit a narrow latitudinal extent (10's to 100's of km) and relatively less longitudinal variation over 100's or 1000's of km (*Knudsen et al.*, 2001). This spatial anisotropy is one motivation for using a meridian scanning photometer (MSP, see §2) to measure auroral luminosity as a sequence of latitude profiles (keogram). As shown in Figure 2, this data can also be used to identify other non-auroral features such as clouds and stars.

## 1.2 Instrument Calibration

Optical designs can be modelled very precisely with modern software tools, but instrument calibration provides essential information about the actual performance. System response is not necessarily constant, but can change either gradually (eg.





**Figure 2.** Keogram from meridian scanning photometer operating at Gillam during the night of December 09 2013 from 06 UT (local midnight) to dawn. Contrast enhancement was applied to emphasize celestial sources; these are circles and ellipses most apparent in the lower half of the image (zenith to southern horizon). Jupiter is the bright feature near scan number 1050 and step number 310.

filter bandpass drift, decreased detector sensitivity) or abruptly (eg. damage during shipping). Such problems could be identified with calibration of instruments in the field. This process must be completely automatic, as many remote sites do not have full-time technical staff. It should be repeated frequently in order to identify abrupt changes in system response, but without interrupting or degrading normal data acquisition. A regular schedule of measurements with portable low-brightness sources

5 (LBS) might satisfy some of these requirements, but would involve a substantial allocation of resources for repeated site visits.

In this report we examine some of the strengths and limitations of astronomical calibration for auroral instruments. We focus on issues related to field cross-calibration of MSPs which have been used extensively for auroral research (see §2 for details).





However, many of these topics can also be applied more generally to other instruments used to study the optical aurora, such as all-sky imagers (ASIs).

A single ground-based instrument may measure photons with wavelengths $\lambda$ arriving from angular locations $\theta, \phi$. External luminosity $I$ is convolved with the instrument response function $f$ to product a measurement $M$ with error $M_\epsilon$

$$M(\theta, \phi, \lambda) = f(\theta, \phi, \lambda) \star I(\theta', \phi', \lambda') + M_\epsilon(\theta, \phi, \lambda) \tag{1}$$

For an ideal device $f$ would be a delta function and $M = I$, but any real measurement will have limited resolution. The goal of calibration or characterization is to determine the instrument response function $f$ in order to better understand the "true" source properties.

The general response function in Equation 1 can be separated into a product of geometric sensitivity $f_G$ and spectral sensitivity $f_S$

$$f_G(\theta, \phi) \times f_S(\lambda) \tag{2}$$

This approximation is not always valid (eg. wide-angle optics coupled to a narrow-band interference filter) but can be usefully applied to many auroral instruments. For convenience we introduce relative response functions ($\hat{f}$) that are normalized to a maximum of one, and combine all scaling into a single system constant $\mathcal{C}$

$$\mathcal{C} \times \hat{f}_G(\theta, \phi) \times \hat{f}_S(\lambda) \tag{3}$$

We show that using Jupiter for field calibration of MSPs provides detailed knowledge about $\hat{f}_G(\theta, \phi)$, estimates of $\mathcal{C}$ that are comparable to darkroom calibration, and useful information about relative spectral response $\hat{f}_S(\lambda)$ at different wavelengths.

### 1.2.1 Geometric

Calibration for auroral instruments with moderate ($\sim 1°$) angular resolution can be achieved using point-like sources located sufficiently far from the entrance aperture. Angular response can be measured by either moving the source or rotating the instrument. The effective field-of-view (or "beam shape") is often azimuthally symmetric around an optical axis with angular polar coordinates $\theta_0, \phi_0$, in which case relative response can be expressed in terms of off-axis angle $\gamma$

$$\hat{f}_G(\theta, \phi) \approx \tilde{f}_G(\gamma; q_1, .., q_N) \tag{4}$$

and some set of instrument parameters $q_i$ (eg. full-width half-max).

Ideally, each instrument would arrive at a field site in exactly the same condition as it left the darkroom. It would be operated exactly as intended (ie. perfectly level and aligned North/South) without changes for the entire design lifetime. In practice it may be difficult to achieve desired alignment to better than a few degrees. The initial orientation may subsequently drift to some more stable state over months or years, or can change abruptly as new instruments are installed nearby. In general, the rotation matrix $R$ required to properly transform from device to local coordinates (eg. azimuth & zenith angle) must be updated regularly in order to ensure that data are scientifically useful.





Determining Euler angles and geometric response model parameters in the field is relatively straightforward for auroral instruments that can detect and resolve at least a few of the brightest stars. Accurate GNSS site location and measurement timing can be combined with astronomical catalogs to predict the local orientation of each star. These can be converted into device coordinates and used to calculate observable quantities such as transit time and zenith angle. Discrepancies between predictions and observations can be minimized to determine optimal parameter values. A single night of good data may be sufficient to achieve sub-degree accuracy, which is adequate for many auroral instruments.

Although stars are essentially point sources at infinity, other immutable properties (eg. location, apparent motion, spectral radiance) may make them somewhat less tractable than darkroom calibration sources. Any given object will not always be visible in the night sky or pass through any specific location in an instrumental field of view. However, a substantial amount of useful information can be gathered over several days or months.

### 1.2.2 Spectral

The relative spectral response of an instrument is essential for quantitative multi-wavelength analysis, such as estimating precipitation energy (*Rees and Luckey*, 1974; *Strickland et al.*, 1989). Spectral response can be most effectively determined with a monochromatic source

$$\int d\lambda' \, \hat{f}_s(\lambda') \, \delta(\lambda - \lambda') = \hat{f}_s(\lambda') \tag{5}$$

that can scan through the wavelength range of interest. For narrow-band devices it may be sufficient to observe a broad-band source $S(\lambda)$ with known absolute spectral flux density. If the source flux is roughly constant near some wavelength $\lambda_j$ for each device channel

$$\int d\lambda \, \hat{f}_s(\lambda) \, S(\lambda) \approx \bar{S}(\lambda_k) \int d\lambda \, \hat{f}_s(\lambda) = \bar{S}(\lambda_k) \, \Delta\lambda_k \tag{6}$$

then the throughput for each channel may be expressed in terms of the effective bandwidth $\Delta\lambda_k$.

Measurements of an absolutely calibrated low-brightness source (LBS) provide estimates of the differential sensitivity to a continuum source characterized in terms of Rayleighs per nanometer. For discrete emission lines the effective bandwidth is also required in order to determine the sensitivity to brightness as expressed in Rayleighs. The equipment necessary for comprehensive calibration (eg. LBS and monochromator) is not always available at remote field sites, so different methods must be established. Many stellar sources provide spectra which are apparently broad-band at typical auroral instrument resolutions on the order of 1 nm. Only relatively bright stars may be above the detection threshold, and absolute flux calibrated spectra are not available for all sources. Still, in certain cases it may be possible for astronomical calibration to produce accurate and repeatable estimates of differential sensitivity.

There does not appear to be a corresponding strategy to determine effective bandwidth in the field. Most stellar spectra are essentially constant in time, so individual sources cannot be used to determine a fixed instument response. Combining many different spectra might in principle allow us to distinguish between changes in effective bandwidth and total sensitivity. However, this would require nearly simultaneous observation of multiple absolutely calibrated sources with different spectral types. Low signal levels might also limit the accuracy of any estimates.




For this study we proceed under the assumption that absolute spectral response cannot be independently determined in the field using only astronomical sources. We presume that normalized transmission integrated across each pass-band

$$\int d\lambda \, \hat{T}(\lambda) \equiv \Delta_\lambda \qquad \hat{T}(\lambda) = [0,1] \tag{7}$$

can be obtained in some other way, and acknowledge that simultaneous changes across multiple channels may not be detected

using methods considered here. For these reasons, we shall tend to focus on the differential calibration coefficient $\dot{\mathcal{C}}$ which can be determined using only astronomical methods. This quantity can also be directly compared to the results of darkroom calibration with an LBS. For auroral studies data numbers $\mathcal{D}$ must be converted to Rayleighs $\mathcal{R}$, and effective bandwidth is required in order to calculate $\mathcal{C}_{\mathcal{R}/\mathcal{D}}$.

### 1.2.3 Photometry

A point source with total power output (radiant flux) $P_0$ and isotropic radiant intensity will produce radiance $S$ which falls off as distance squared.

$$S = \frac{P_0}{4\pi R^2} \qquad \text{watt} \cdot \text{meter}^{-2} \tag{8}$$

An observer at some distance $r$ will intercept an amount of power

$$P_\delta = S \, A_{\text{eff}} \tag{9}$$

proportional to the effective receiver surface area $A_{\text{eff}}$ .

Power from an extended source can be expressed in terms of a volume emission rate $\rho(r,\theta,\phi)$ integrated over the entire source region weighted by the receiver angular sensitivity $G(\theta,\phi)$

$$P_V = \oiint d\Omega \frac{L\,G}{4\pi} \qquad L \equiv \int_0^\infty dr\, \rho(r) \tag{10}$$

where the radial integral $L$ has units of radiance ($\text{watt} \cdot \text{meter}^{-2} \cdot \text{sr}^{-1}$) and is often referred to as the "column emission rate".

For a uniform source radiance the total received power

$$P_V = \oiint d\Omega \frac{L(\theta,\phi)}{4\pi} A_{\text{eff}} \hat{G}(\theta,\phi) \approx L \, A_{\text{eff}} \, \Omega_0 \tag{11}$$

depends on the product of the effective area and the effective solid angle. When signals are detected from some point source, we might ask what equivalent volume emission would produce the same observed power. For a uniform emission region the result

$$P_\delta = P_V \qquad \rightarrow \qquad L = \frac{S}{\Omega_0} \tag{12}$$

depends only on the effective solid angle.





Auroral intensities are customarily expressed in units of Rayleighs (*Hunten et al.*, 1956; *Baker*, 1974; *Baker and Romick*, 1976; *Brändström et al.*, 2012)

$$4\pi L_\gamma(\lambda) \equiv \mathcal{I}(\lambda) \qquad 10^{10}\ \mathrm{photon\cdot s^{-1}\cdot m^{-2}} \tag{13}$$

where the subscript $E$ indicates energy flux and $\gamma$ is photon number flux. For narrow-band channels

$$\mathcal{I}(\lambda) = \int \dot{\mathcal{I}}(\lambda) \approx \dot{\mathcal{I}}\,\Delta\lambda = 4\pi \frac{\dot{S}_E}{\Omega_0} \frac{\lambda}{hc}\Delta\lambda \tag{14}$$

converting differential radiant spectral density $\dot{S}$ to equivalent Rayleighs per nanometer $\dot{\mathcal{I}}$ requires only the effective solid angle, which can also be estimated from observations of a point source. Working with Rayleighs requires some additional knowledge in the form of the effective bandwidth $\Delta\lambda$. As this is also true for darkroom LBS calibration, we focus here on relating $\dot{I}$ in Rayleighs per nanometer to $\dot{S}$ in Watts per metre-squared per nanometer.

## 1.3  Astronomical sources

Extra terrestrial objects have many properties which are required for accurate calibration. Locations in the celestial sphere are known to arc-second resolution or better, which is more than enough to determine the orientation and geometric response of auroral instruments. Absolute spectral irradiance profiles are available for many sources, providing opportunities for photometric calibration of narrow-band instruments. Total visible intensity of most sources is essentially constant, allowing for long term monitoring of system performance. A single object can be viewed simultaneously by multiple instruments at nearby sites, facilitating quantitative inter-comparisons.

Most astronomical objects are effectively point sources, which is convenient for geometric calibration, but can introduce complications for auroral instruments designed to observe extended emission regions. Only the brightest stars can produce count rates comparable to background contributions such as airglow. Celestial source brightness spans a wide range and is usually expressed in terms of logarithmic magnitude $m$

$$I = \sqrt[5]{100}^{\,m} \approx 2.512^m \tag{15}$$

so that the relative intensity of two sources can be determined from the difference of their magnitudes. Absolute flux distributions as a function of wavelength are available for most of the brightest stars, including Vega (*Colina et al.*, 1996), Sirius (*Bohlin*, 2014), and Arcturus (*Blackwell et al.*, 1975; *Griffin and Lynas-Gray*, 1999). Other catalogs contain many other stars (*Hayes*, 1985; *Alekseeva et al.*, 1996, 1997; *Bohlin*, 2007, 2014), but the majority may be too dim for reliable observation by typical auroral instruments.

Conversely, the sun is so bright that direct observation will saturate detectors designed for relatively faint aurora. *Thuillier et al.* (2003) provide an absolutely calibrated distribution of flux versus wavelength at 1 AU with sub-nanometer spectral resolution. For a nominal instrument solid angle of 2 milli-steradians (3° of arc) the apparent solar brightness at 556 nm is roughly 3 teraRayleighs per nanometer (Table 2). Daytime operations are only possible for systems that respond to an extremely narrow range of wavelengths (*Galand et al.*, 2004).



**Table 2.** Selected astronomical source irradiance at Earth. Rayleighs are for a viewing solid angle of $\Omega = 0.002$ steradians (2.9° of arc).

| | [nm] | $[s\ m^2\ nm]^{-1}$ | | [R / nm] |
| | | [J] | [#] | |
| --- | --- | --- | --- | --- |
| jupiter | 486 | 4.78e-10 | 1.17e+09 | 735 |
| jupiter | 556 | 5.45e-10 | 1.53e+09 | 958 |
| sirius | 556 | 1.35e-10 | 3.78e+08 | 237 |
| vega | 556 | 3.44e-11 | 9.63e+07 | 60.5 |
| moon | 556 | 4.63e-06 | 1.3e+13 | 8.14e+06 |
| sun | 556 | 1.81 | 5.07e+18 | 3.18e+12 |



**Figure 3.** Spectra of solar irradiance (upper curve) (*Thuillier et al.*, 2003) and Jupiter albedo (*Karkoschka*, 1998). Inset contains same quantities in visible wavelength range.



Although direct sunlight is unsuitable as a calibration source for most auroral instruments, scattering from other bodies in the solar system can provide more reasonable levels of brightness. The irradiance of an arbitrary body $x$ can be modeled by isotropic emission from the sun incident on a sphere with radius $R_x$ at distance $D_{Sx}$, followed by scattering and absorption leading to some fraction of flux travelling a distance $D_{xE}$ to arrive at the top of Earth's atmosphere. We can group terms that depend on wavelength and time into $B(\lambda)$ and $A(t)$ respectively

$$I_{xE}(\lambda, t) = A(t) \times B(\lambda) \tag{16}$$

where the solar power $P_s(\lambda)$ and planetary albedo $\epsilon$ are both assumed to be time independent to 1% or less. We may express irradiance in terms of the total solar irradiance (TSI $\sim 1360$ watts/metre-squared) at a fixed distance of 1 AU.

$$B(\lambda) \equiv TSI(\lambda)\,\epsilon(\lambda) \tag{17}$$

A phase correction term $\Phi(\varphi)$ accounts for any non-Lambertian scattering as a function of angle $\varphi$ between illumination and observer.

$$A(t) \equiv \frac{R_x^2\, D_{SE}^2}{D_{Sx}^2\, D_{xE}^2}\, \Phi(\varphi)\cos(\phi) \tag{18}$$

For example, illumination from a full moon ($\phi = 0$) is reduced by a factor of 3e-6 (m $\sim$ 14) relative to direct sunlight. Despite this substantial decrease, the equivalent brightness of roughly one megaRayleigh per nanometer (Table 2) is still a hundred times brighter than the brightest aurora. For many instruments the angular size of the moon is neither point-like nor beam-filling, requiring careful attention to details such as wavelength dependent albedo varying across the disk (*Kieffer and Stone*, 2005), and making phase calculations more complicated. For these reasons, the moon is not commonly used for calibrating auroral instruments.

After the moon, Jupiter is currently the brightest celestial object that can be regularly observed well past astronomical twilight. Peak visible magnitude is nearly four times Sirius (the brightest star), making Jupiter easy to identify in the night sky. A detailed spectral distribution of Jupiter's albedo is given by *Karkoschka* (1998). This can be combined with the solar spectrum of *Thuillier et al.* (2003) to predict the wavelength dependence of reflected light given in Table 3.

Other bodies in our solar system are less suitable as calibration sources. Mercury is only visible from Earth during the daytime when looking near the sun. Venus can often be seen near dawn or dusk, but always with excessive amounts of indirect sunlight. Mars can be visible at night for several months in a row, but this ideal configuration only occurs on alternate years. (Figure 5). Albedo can vary considerably during dust storms and a wide range of $\varphi$ means that the phase function $\Phi$ must be very precisely determined (*Mallama*, 2007). Saturn is roughly one-tenth as bright as Jupiter, with complex albedo variations due to ring geometry ($V = -0.62$ to $+1.31$) (*Mallama*, 2012). The remaining outer planets are simply too dim for reliable detection by most auroral instruments.

As Jupiter and Earth each orbit around the Sun, their relative motion produces significant variations in apparent magnitude and position as shown in Figure 4. In recent years Jupiter and the Earth have been closest during winter in northern hemisphere,





maximizing brightness during the optimal period for observations with auroral instruments. As shown in Figure 5, Jupiter transit at Gillam Manitoba currently occurs near sunrise in early October and sunset in February. An orbital period of 11.89 years means that opposition will advance by roughly one month per year. Optimal configurations with transit near midnight during northern winter started in 2011, will continue until 2016, and then begin again in 2022. Previous windows of opportunity include 1988-1993 and 1999-2005. Any historical data acquired during these years could conceivably be retrospectively calibrated using Jupiter.

**Figure 4.** Jupiter as seen from northern auroral zone from 2009 to 2014. [1] Top: apparent visual magnitude (negative is brighter). Different curves correspond to results from older references ($V(1,0)$ =-9.25), newer references (-9.40) and calculations in this study (-9.426). Middle: declination, which is effectively the same for any terrestrial observer (parallax$\approx 0$). Bottom: relative air mass for transit at Fort Smith, Gillam, Athabasca, and Pinawa.

During in this study we identified a systematic difference between our flux calculations for Jupiter and the corresponding magnitude value provided by widely available astronomy software (*Downey*, 2015) using the formula

$$V = V(1,0) + 5\log_{10}(dr) + \Delta m(i) \tag{19}$$



where $V(1,0)$ is the magnitude at 1 AU with $i = 0$ and $\Delta m(\phi)$ is the magnitude phase correction. Our results were calculated by entering standard distances into Equation 18 with irradiance and reflection from *Thuillier et al.* (2003) and *Karkoschka* (1998). We obtained equivalent values of $V(1,0) \approx -9.426$ that were nearly 20% larger than the standard result of $V(1,0) = -9.25$. Eventually we discovered that the widely used lower value came from the 2nd edition of the Explanatory Supplement to the Astronomical Almanac (*Seidelmann*, 1992) but the most recent 3rd edition (Table 15.8 *Seidelmann*, 2005) now indicates $V(1,0) = -9.40$, which differs from our results by only 2%. This exemplifies the level at which we were able to cross-check our results against other references. It also demonstrates that even astronomical "constants" may be a work in progress.

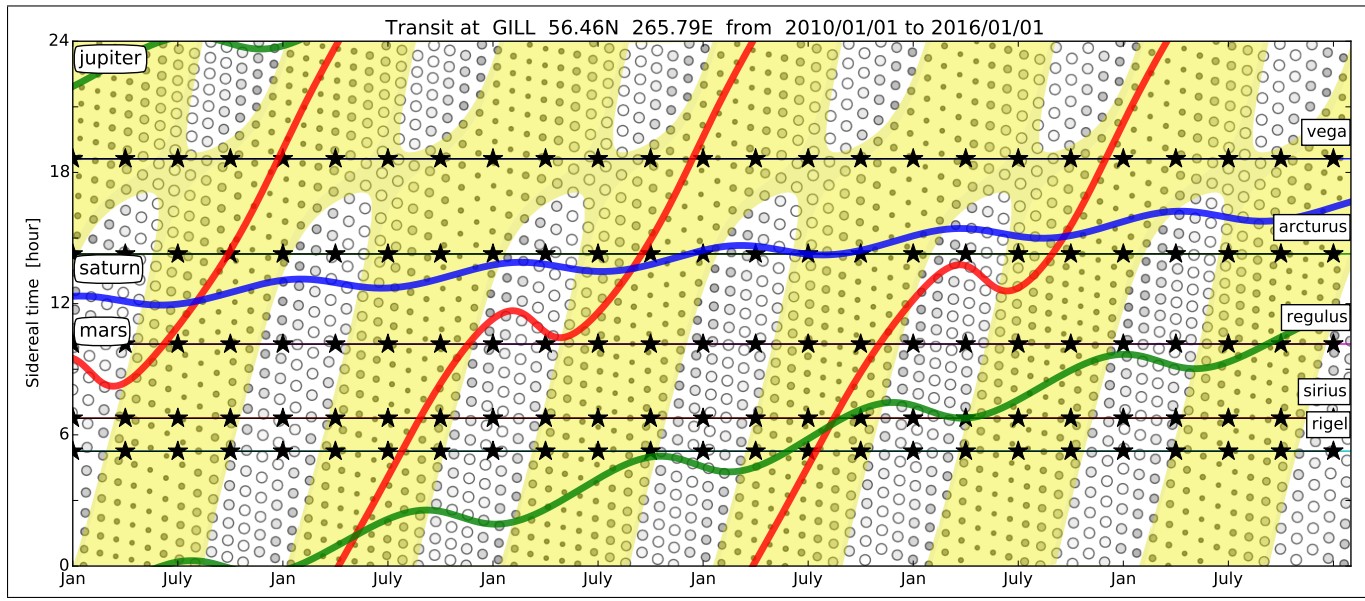

**Figure 5.** Planetary right ascension over time indicated by thick colored lines. Stars indicated by thin black lines remain at constant RA. Yellow contours correspond to daytime between nautical sunrise and sunset ($6°$ below horizon). Size of small circles are proportional to lunar phase with yellow indicating daytime.

## 1.4 Atmospheric effects

Light arriving at the top of the Earth's atmosphere may undergo significant changes by the time it arrives at a ground-based observer. Gradients in the refractive index will bend ray paths, changing the apparent arrival angle. The magnitude of this effect increases with zenith angle but is only on the order of 5 arc-minutes at $10°$ elevation above the horizon. This might be important for astronomical applications, but is negligible for most optical auroral devices with precision requirements on the order of $1°$.

In contrast, variations in atmospheric transmission can be important even at moderate zenith angles. Atmospheric scattering and absorption processes will reduce the radiant flux detected by a ground-based observer (*Sterken and Manfroid*, 1992). The





decrease in apparent magnitude can be modelled as

$$\Delta m(\lambda, \zeta) = \kappa(\lambda)\, X(\zeta) \tag{20}$$

where the relative air mass $X$ as a function of zenith angle $\zeta$

$$X(\zeta) \approx 1 + (1 - c_1)\,Z - c_2 Z^2 - c_3 Z^3 \qquad Z = \frac{1 - \cos\zeta}{\cos\zeta} \tag{21a}$$

is equal to one at the zenith (ie. $X(0) = 1$) and increases by a factor of 5 at $10°$ elevation above the horizon (*Tomasi and Petkov*, 2014). For convenience we may separate zenith angle and wavelength effects

$$E(\lambda, t) = \mathrm{E_k}(\lambda)^{X(t)} \qquad \mathrm{E_k} \equiv 2.512^{-\kappa(\lambda)} \tag{22a}$$

where $\mathrm{E_k}$ is the transmission through one standard air-mass (ie. at zenith).

Empirical results from several nights of astronomical observations near sea level (*Vargas et al.*, 2002) show total extinction
ranging from $\kappa = 0.312 - 0.604$ and $\kappa = 0.180 - 0.347$ for standard blue and red filters respectively. *Zhang et al.* (2013) found $\kappa_g = 0.69$ and $\kappa_r = 0.55$ at a low altitude (170m) high humidity location. *Tomasi and Petkov* (2014) present an extensive review of optical airmass properties for the Arctic and Antarctic.

For this study we use values from *Patat et al.* (2011) to provide a lower bound on extinction effects. The upper bound is estimated using an empirical model based on *Vargas et al.* (2002) and *Zhang et al.* (2013).

**Table 3.** Spectral variation of solar irradiance at Earth (*Thuillier et al.*, 2003), albedo of Jupiter (*Karkoschka*, 1998), and atmospheric extinction at Cerro Paranal (*Patat et al.*, 2011). Column 5 is the product of solar irradiance at 1AU and Jupiter albedo (defined as $B(\lambda)$ in Equation 18) with units of watts per metre-squared per nanometre. Atmospheric transmission $\mathrm{E_k}$ at zenith is related to extinction $\kappa$ by Equation 22a. Column 8 is the product of solar irradiance, Jupiter albedo, and atmospheric transmission with units of watts per metre-squared per nanometre.

| wavelength | Solar flux [$\mathrm{m^{-2}\,nm^{-1}}$] | | Jupiter | $B(\lambda)$ | atmosphere | | $B(\lambda)\,\mathrm{E_k}(\lambda)$ |
|---|---|---|---|---|---|---|---|
| [$nm$] | [photon $s^{-1}$] | [W] | albedo | [$W\,\mathrm{m^{-2}\,nm^{-1}}$] | $\kappa$ | $\mathrm{E_k}$ | [$W\,\mathrm{m^{-2}\,nm^{-1}}$] |
| 470.9 | 1.783E+018 | 2.004 | 0.446 | 0.893 | 0.187 | 0.842 | 0.752 |
| 480.0 | 1.951E+018 | 2.096 | 0.454 | 0.952 | 0.179 | 0.848 | 0.807 |
| 486.1 | 1.701E+018 | 1.788 | 0.455 | 0.814 | 0.171 | 0.854 | 0.695 |
| 495.0 | 2.027E+018 | 2.005 | 0.470 | 0.942 | 0.160 | 0.863 | 0.813 |
| 557.7 | 2.315E+018 | 1.799 | 0.515 | 0.927 | 0.127 | 0.889 | 0.824 |
| 625.0 | 2.310E+018 | 1.627 | 0.495 | 0.805 | 0.101 | 0.912 | 0.734 |
| 630.0 | 2.481E+018 | 1.646 | 0.520 | 0.855 | 0.097 | 0.915 | 0.782 |

### 1.4.1   Transit zenith angle

Zenith angle at transit depends on the observer latitude $\Lambda$ and declination of the source. Consequently, two observers viewing the same source from different latitudes will be looking through different air masses. This can produce systematic differences



in brightness of a few percent or more depending on the latitude offset $\Delta\Lambda$ and extinction $E_k$

$$I_2/I_1 \propto E_k{}^{\Delta X} \qquad \Delta X \approx \frac{1}{\cos(\zeta_1 + \Delta\Lambda)} - \frac{1}{\cos(\zeta_1)} \tag{23}$$

Calibration using Jupiter (or any other planet) will be further complicated by corrections for varying declination. Figure 4 shows several years variation of air mass for Jupiter transit at the four field sites considered in this study. A significant transition occurs between large latitude dependent extinction before 2011 to relatively uniform low levels afterward. The effects for this study are only on the order of a few percent, but are clearly evident in results presented in §3.3. This provides some assurance that our analysis procedures are accurate near the 1% level. Of course, calculating the effects of varying declination requires atmospheric extinction coefficients that may not be very well known. This is a challenge, but also an opportunity to test which extinction models produce the best agreement with observations.

Declination differences can even alter the intensity ratio between two different wavelengths (heterochromatic extinction in *Sterken and Manfroid* (1992))

$$I_2/I_1 \propto \Delta E_k{}^{X(\zeta)} \qquad \Delta E_k \equiv 2.512^{\kappa(\lambda_1) - \kappa(\lambda_0)} \tag{24}$$

because extinction is a non-linear function of air mass. This effect is considered in §3.4 and found to be significant.

## 2 Meridian Scanning Photometer

Auroral luminosity is often spatially anisotropic, with latitude structuring on scales of 1-100 km and longitudinal features extending from 100s up to 1000s of kilometres. Consequently, some instruments are designed with reduced azimuthal coverage in exchange for improved sensitivity along a latitude profile. These systems may be referred to as meridian imaging spectrographs (MIS) or meridian scanning photometers (MSP) depending on the technology used for spectral discrimination and photon detection. In this paper we explore issues related to field cross-calibration of a specific MSP design that has been used extensively for auroral research in Canada. Many of these topics can also be applied more generally to other auroral optical devices.

Data used for this study were obtained from a network of four multi-spectral auroral meridian scanning photometers. These systems were based on the meridian scanning photometer array (MPA) component of the CANOPUS project (*Rostoker et al.*, 1995) which operated MSPs at three sites in a latitude chain: Rankin Inlet, Gillam, Pinawa (the "Churchill line"), and a fourth auroral zone site two hours to the west in Fort Smith. The primary goal was to detect proton aurora at 486.1 nm and electron aurora at several wavelengths (see Table 4) in order to determine precipitation species, characteristic energy, and energy flux. The array was operated continuously for nearly 20 years, producing a large high-quality data set which was the foundation for important research on topics including substorms (*Samson et al.*, 1992), the polar cap boundary (*Blanchard et al.*, 1995, 1997), poleward boundary intensifications (*Lyons et al.*, 1999; *Zesta et al.*, 2000), and the B2i isotropy boundary (*Donovan et al.*, 2003).

Due to bandwidth limitations, most raw instrument output was down-sampled by averaging in space and time in order to produce a uniform data stream for real-time transmission. Full high resolution data were available over a serial "campaign



port". In later years, data loggers were used at some sites to record the full resolution data; several years of "high-res" MSP data are available for retrospective re-calibration. The more extensive "low-res" dataset is averaged into 17 latitude bins per scan, which is adequate for auroral science, but diminishes the ability to resolve elevation from individual star transits.

The original CANOPUS MSPs were built by an industrial contractor (*Johnston*, 1989) based on a series of instruments developed at the National Research Council of Canada (NRCC). Calibration of the prototype was carried out in 1985 by NRCC and the University of Saskatchewan; the results of which led to several design modifications. The first field system was commissioned at Gillam in February 1986, with all four units operational by early 1988. By the late 1990's it was increasingly obvious that the instruments were nearing end-of-life. The primary concern was the mirror motors which had driven several billion steps, but many other issues (eg. data acquisition, high voltage supplies, photomultiplier tubes) were also causing problems. Eventually, a lack of spare parts resulted in significant failures and data loss.

An MSP revitalization project was carried out at the University of Calgary starting in 2007. The goal was to provide replacement systems with equivalent functionality. System design was based closely on the original instruments in order to minimize risk, with legacy mechanical and optical components reused where possible. Initial development was carried out on the legacy system at Rankin Inlet which was broken beyond repair. The detector was replaced with a new PMT, high voltage supply, and pulse-counting circuit. Anti-reflection coatings were added to several optical elements, with system throughput optimized with predictions from optical modelling software and confirmed with quantitative testing. All of the old filters were replaced, as was the filterwheel motor. The scanning mirror assembly was upgraded to provide $0.09°$ elevation steps (4000 steps per $360°$). Thermal and power control systems were completely replaced. An FPGA coordinates for low level timing and synchronization, while a Linux PC-104 was responsible for data acquisition and overall system control.

After darkroom calibration and local field trials the new prototype system was deployed at Gillam and operated adjacent to the legacy system which was still functioning intermittently. The original Gillam system was then upgraded and sent to Fort Smith (2009), the old Fort Smith system upgraded and installed at Pinawa (2010), and the old Pinawa system upgraded and moved to a new site near Athabasca (2011). Additional improvements were implemented in later systems, motivating a round of upgrades in 2012 to the Gillam and Fort Smith units. The entire rebuild process took more than four years and involved multiple personnel at the University of Calgary. Despite careful attention to tracking changes, there are still some functional differences between the first and last refurbished systems. Many of these issues have been identified with internal calibration procedures, but astronomical sources provide useful insight about comparative instrument performance.

The new Calgary MSPs use the same filter-wheel design as CANOPUS to acquire data from eight spectral channels, with 486.1 nm duplicated in order to increase SNR for faint proton aurora. Accurate photometry of rapidly varying aurora requires effectively simultaneous measurements of background and signal. This is accomplished by rotating the filter-wheel at 1200 RPM (20 Hz) and gating the detector to provide successive 12.5 ms sample spacing. Some details about filter sequencing is given in Table 4; for simplicity all subsequent multi-channel data will presented in wavelength order (blue to red).

Interference filter transmission and blocking as a function of wavelength were provided by the manufacturer and summarized in Table 5. Results were very close to specifications (FWHM 3 nm for the blue filters and 2 nm for the others). Transmission peaks were broad and flat with maxima around 80%, which is the key parameter for optimizing detection of narrow emission



**Table 4.** MSP filter wheel sequence.

| wavelength | description | filter wheel position | |
| --- | --- | --- | --- |
| [nm] | | CANOPUS | Calgary |
| 470.9 | $N_2^+$ energy flux | 5 | 6 |
| 480.0 | blue background (1) | 1 | 2 |
| 486.1 | $H_\beta$ (1) | 2 | 3 |
| 486.1 | $H_\beta$ (2) | 3 | 4 |
| 495.0 | blue background (2) | 4 | 5 |
| 557.7 | OI "green-line" | 6 | 7 |
| 625.0 | "red-line" background | 7 | 0 |
| 630.0 | OI "red-line" | 0 | 1 |

lines. The effective passband

$$\Delta\lambda_j = \int d\lambda \, \hat{T}_j(\lambda) \tag{25}$$

is the relevant quantity for broad-band calibration sources ie. converting from Rayleighs per nanometer to Rayleighs. These data suggest typical passband and transmission variations on the order of 5% between different sets of filters.

**Table 5.** Characteristics of three sets of nominally identical narrow band filters. Passband is integral of transmission profile, 90% bandwidth is the range between 5% and 95% points of the cumulative transmission.

| [nm] | passband [nm] | | | peak transmission [%] | | | 90% bandwidth [nm] | | |
| --- | --- | --- | --- | --- | --- | --- | --- | --- | --- |
| 470.9 | 2.483 | 2.362 | 2.355 | 82.94 | 79.36 | 78.28 | 3.60 | 3.40 | 3.50 |
| 480 | 2.592 | 2.418 | 2.661 | 85.59 | 78.60 | 87.74 | 3.10 | 3.20 | 3.20 |
| 486.1 | 2.605 | 2.587 | 2.615 | 88.26 | 85.95 | 87.57 | 2.90 | 3.00 | 3.00 |
| 486.1 | 2.572 | 2.222 | 2.509 | 84.71 | 74.21 | 83.44 | 3.10 | 3.00 | 3.10 |
| 495 | 2.607 | 2.525 | 2.584 | 88.40 | 85.74 | 87.27 | 3.30 | 3.40 | 3.50 |
| 557.7 | 1.788 | 1.728 | 1.920 | 82.93 | 78.93 | 88.53 | 4.60 | 4.90 | 4.30 |
| 625 | 1.624 | 1.632 | 1.588 | 84.46 | 87.37 | 86.09 | 4.20 | 3.20 | 2.80 |
| 630 | 1.597 | 1.590 | 1.558 | 86.67 | 84.83 | 83.81 | 2.40 | 2.60 | 2.40 |

5      Light which passes through the filters is detected by a photo-multiplier tube (PMT) with photocathode quantum efficiency ranging from 20% at 400 nm to 2% at 750 nm; this response was selected to maximize response for the faint $H_\beta$ emissions. A dynode chain amplifies each electron to produce a cascade which triggers a pulse-counting circuit. The high-voltage power supply required for this process is quite stable over short intervals under ideal conditions, but may change during extended field operations. Photocathode aging and high-voltage drift are likely to be the primary cause of any long-term reduction in system
10    sensitivity.



PMTs dead-time produces a non-linear response at high count-rates. This pulse pile-up effect can be largely removed if the time resolution $\tau$ of the system is known and is not significantly longer than the signal count interval. For the PMTs used in this study nonlinearity only becomes important for count rates greater than $10^5$ photons per second. These rates can be produced by very bright aurora but are not a problem for any astronomical sources except the Sun and Moon.

Meridian scans are achieved with a $45°$ mirror and a stepping motor. Many MSPs rotate the mirror at a fixed rate in order to produce data from evenly spaced elevations. Both the original and refurbished systems instead utilize a sequence of variable steps chosen to produce nearly constant exposure times as a function of linear distance at auroral altitudes. This detail is relevant to this study because Jupiter transit profiles will be measured with different resolution depending on transit elevation. The effects are expected to be small, but must be kept in mind when considering multi-year variability.

## 2.1 System sensitivity

The relationship between incident photon flux $\mathcal{P}(\lambda)$ and measured channel count rate $\mathcal{D}_k$

$$\mathcal{D} = A_{\text{eff}} \, M_x \, \Delta t \int d\lambda \, \mathcal{P}(\lambda) \, T_k(\lambda) \, Q(\lambda) \tag{26}$$

depends on the effective aperture allowing photons into the system ($A_{\text{eff}}$), channel multiplexing efficiency ($M_k$), filter transmission ($T_k$), measurement interval ($\Delta t$), and the detector efficiency $Q(\lambda)$.

For wide-band input through narrow-band filters the process can be written in terms of filter peak transmission $T_k$ and bandwidth $\Delta \lambda_k$

$$\mathcal{D}(\lambda_i) \approx \mathcal{P}(\lambda_k) \, A_{\text{eff}} \, M_k \, \Delta \lambda_k \, T(\lambda_k) \, \Delta t \, Q(\lambda_i) \tag{27}$$

giving a response $_k\mathcal{C}_{\mathcal{D}/\mathcal{P}}$ for each channel

$$_k\mathcal{C}_{\mathcal{D}/\mathcal{P}} = \frac{\mathcal{D}(\lambda_k)}{\mathcal{P}(\lambda_k)}$$

$$= A_{\text{eff}} \, M_x \, \Delta \lambda_k \, T(\lambda_k) \, \Delta t \, Q(\lambda_k) \tag{28}$$

in terms of measured $\mathcal{D}$ and predicted $\mathcal{P}$ for each filter wavelength. We will use the term "calibration coefficient" to refer to differential brightness per count

$$_k\mathcal{C}_{\dot{\mathcal{R}}/\mathcal{D}} = A_{\text{eff}} \, M_x \, T(\lambda_k) \, \Delta t \, Q(\lambda_k) \tag{29}$$

which is the quantity of interest when converting data to physical units. However, we will express subsequent results in terms of the reciprocal "sensitivity"

$$\mathcal{C}_{\mathcal{D}/\dot{\mathcal{R}}} = 1/\mathcal{C}_{\dot{\mathcal{R}}/\mathcal{D}} \tag{30}$$

for which higher numbers are better.


## 2.2 Darkroom calibration

All systems have been calibrated at the University of Calgary using a low brightness source (LBS) with spectral radiance measured by the Canadian Institute for National Measurement Standards. Several sets of calibration results for one instrument at different times are shown in Table 6. Results from two sucessive days (November 21 & 22 2014) agree to 1% or better,

suggesting that the calibration process is highly repeatable. Earlier results from 2010 indicate that the system was about 5% more sensitive in all channels, but with only two measurements over more than 4 years, it is impossible to determine whether this corresponds to a gradual decline or an abrupt change at some time during shipping or field operations.

**Table 6.** Fort Smith MSP channel sensitivity $\mathcal{C}_{\dot{\mathcal{R}}/\mathcal{D}}$ [Rayleighs/nm/count] determined by darkroom LBS calibration.

| Site Date Device | 471.0 | 480.0 | 486.0 | 486.0 | 495.0 | 557.7 | 625.0 | 630.0 |
|---|---|---|---|---|---|---|---|---|
| fsmi 20100112 msp-02 | 0.2712 | 0.2570 | 0.2446 | 0.2474 | 0.2666 | 4.5029 | 0.8188 | 0.8598 |
| fsmi 20141121 msp-02 | 0.2935 | 0.2756 | 0.2647 | 0.2685 | 0.2900 | 5.8267 | 0.9180 | 0.9742 |
| fsmi 20141122 msp-02 | 0.2942 | 0.2735 | 0.2645 | 0.2687 | 0.2904 | 5.9789 | 0.9258 | 0.9833 |

## 3  Data analysis

In this section we present methods for extracting useful calibration information from Jupiter transits in MSP data. There are

five topics organized by which parameter is under consideration and what supporting measurements are required with results that range from precise and absolute to uncertain and relative. Optical field of view is considered in §3.1, device orientation in §3.2, magnitude variation in §3.3, spectral ratios in §3.4, and absolute sensitivity in §3.5.

Each of the MSPs considered in this study executes a sequence of repeated scans from the northern to southern horizon. Every scan consists of multiple steps through a $160°$ elevation range, with measurements acquired through multiple filters at

each step. The resulting data stream has units of "counts" or simply "data numbers" ($\mathcal{D}$) and can be represented by a $[K, M, N]$ array of 16-bit numbers where $K = 8$ is the number of filters, $M = 544$ is the usual number of elevation steps for the rebuilt MSPs, and $N = 120$ scans are acquired during each hour (30-second cadence).

Ephemeris software (*Downey*, 2015) was used to calculate the time and elevation corresponding to the transit of Jupiter through the local meridian containing the zenith and terminated by the celestial poles. To start we assumed that instruments

were perfectly level and had azimuths pointing directly north in order to obtain a starting point for identifying actual transits. A keogram sub-region centered on the predicted transit was used to fit a two-dimensional generalized Gaussian model

$$
\mathcal{D}(x, y) = D_0 \, \exp\left[ -\left| \frac{\bar{x}}{\alpha_x} \right|^{\beta_x} - \left| \frac{\bar{y}}{\alpha_y} \right|^{\beta_y} \right] \tag{31}
$$
$$
+ B_0 \left\{ 1 + B_x \bar{x} + B_y \bar{y} + B_{xy} \bar{x} \bar{y} \right\}
$$





**Figure 6.** Gillam MSP observations of Jupiter on November 22, 2011. Shading in central panel corresponds to counts for each scan and step (higher DN are darker, ranging from 0 to 1500 DN), contours indicate best fit 2D Gaussian. Right panel is elevation profile obtained by averaging over time (symbols) and best fit Gaussian (dotted line). Top panel is time profile obtained by averaging over elevation. Dashed lines indicate the predicted transit time (off scale) and elevation for ideal north-south scan.

where $D_0$ and $B_0$ are signal and background, $\bar{y} = y - y_0$ and $\bar{x} = x - x_0$ are the elevation and time relative to the transit peak $x_0, y_0$, $\alpha_{x,y}$ are profile widths, and $\beta_{x,y}$ are scaling parameters. Jupiter transit profiles were initially modelled with a simpler bivariate Gaussian ($\beta_x = \beta_y = 2$) which could usually achieve model/data differences on the order of 10%. The more general representation in Equation 31 was introduced in an attempt to ensure that model error would not be a limiting factor for analysis at the 1% level. We subsequently found that the coefficients also provided a useful measure for classifying transit quality, and more clearly identified minor azimuthal asymmetry in the optical response.



**Figure 7.** Jupiter transit time (UT) observed at Gillam during 2011-2014 northern hemisphere winters. Each symbol corresponds to a single night, large symbols to higher-quality fits.

The polynomial background model is effective for mitigating effects from dawn/dusk gradients and scattered moonlight. This significantly increases the number of transits which could be used for estimating orientation and field-of-view, although relatively few of these additional events are suitable for photometric calibration. Figure 7 shows Gillam transit times obtained over three winters. Sequences of good transits correspond to cloudless nights, gaps to periods of poor visibility near full Moon.





### 3.1 Field of view

Stars and distant planets are effectively point sources when viewed with a single pixel detector (PMT) through optics with angular resolution on the order of $1°$. Each MSP elevation sweep over an astronomical source will produce a profile that corresponds to the "vertical" optical angular response. Similarly, a time sequence of observations from a fixed elevation should

provide a complementary measure of "horizontal" optical beam shape. This is illustrated in Figure 6 with a full two-dimensional (elevation and time) distribution of observed counts along with corresponding elevation and time profiles. Each profile is approximately Gaussian, and the combined two dimensional pattern is fairly well modelled by the bivariate generalized Gaussian in Equation 31. A complete transit profile extends over 10 minutes, during which time viewing conditions may change considerably. In contrast, each elevation sweep over Jupiter lasts for only a few seconds.

Fitted horizontal (time) and vertical (elevation) beam widths from the Gillam MSP are plotted in Figure 8. There is a cluster of points near $\sigma \sim 1.1°$ that presumably corresponds to the actual beam shape. Other points are generally associated with sub-optimal viewing conditions (eg. clouds or aurora). Seasonal average estimates of horizontal and vertical beam width for Gillam and Fort Smith sites are presented in Table 7. Results are consistent with all instruments having similar horizontal and vertical widths: $\sigma \approx 1.07°$ (FWHM $\sim 3.0°$). Average beam widths have standard deviations less than $0.05°$ and standard errors less

than $0.01°$; typical beam solid angles are approximately $2.30 \times 10^{-3}$ steradians with uncertainties of a few percent.

The effective solid angle $\Omega_0$ is essential for comparing flux from distant point sources to distributed auroral emissions. For several years of Fort Smith data the average value was 2.07 milliSteradians with standard deviation of 0.12, and standard error of the mean less than 1%.

### 3.2 Orientation

An ideal MSP would be aligned to produce scans with predetermined azimuth and elevation. For outdoor installations at remote field sites it can be difficult to reduce leveling errors below a few degrees. Further complications may arise as the ground freezes in autumn and thaws in spring. Geographic azimuth may be difficult to precisely determine unless a detailed site survey is available. Alignment with magnetic north can also be challenging unless the site is magnetically clean and there are no geomagnetic disturbances. Over longer periods the magnetic declination may change significantly (see Table 1) due to

secular variation in the geomagnetic field.

Fortunately, it is possible to accurately determine instrument orientation from transit observations. Starting with site locations obtained using GPS, observed transit times were used to calculate the actual elevation and azimuth of Jupiter for each night. These were interpreted in terms of two device angles. First, azimuth offset was attributed to horizontal orientation of a level instrument. Second, the difference between nominal mirror elevation and actual target elevation was attributed to instrument

"tilt" from level.

Results for three seasons at Gillam are shown in Figure 9. Azimuth estimates are extremely stable over time, with jitter $< 1°$ and no apparent drift. Tilt estimates from the first two seasons are even less variable, although there appears to be a small jump in early November. Examination of results from the other three sites (not shown) finds a similar feature at Fort Smith (FSMI),



**Figure 8.** Optical beam width $\sigma$ determined by fitting a generalized Gaussian to observations of Jupiter by an MSP at Gillam over three winters.

a smaller shift at Pinawa (PINA), and no obvious change at Athabasca (ATHA). These results are consistent with "frost heave" occurring in early winter as moisture in the soil freezes. The lack of this effect at ATHA may be be due to better foundations for the instrument platform.

A yearly summary of orientation parameters for each site is presented in Table 7. For cases with 30 or more good transits the standard deviations are less than $1/2°$ and uncertainties in the average (standard errors) are less than $0.1°$. This allows data to be accurately mapped into other coordinates (ie. geographic); even minor changes to instrument alignment can be easily identified.



**Figure 9.** Gillam MSP orientation inferred from observations of Jupiter. Each symbol corresponds to one transit during a single night. Large symbols correspond to "good" beam widths with both vertical and horizontal $\sigma \sim 1°$. Small symbols correspond to all other events.

**Table 7.** Instrument orientation and beam width from all good transits at Gillam and Fort Smith during each winter. Averages and standard deviations in degrees for azimuth, tilt, beam width, beam height. Solid angle average in milli-steradians and percent standard error.

| site | year | N | azimuth | tilt | $\sigma_h$ | $\sigma_v$ | $\Omega$ |
|------|------|---|---------|------|-----------|-----------|----------|
| GILL | 2011-12 | 73 | $6.65 \pm 0.16$ | $0.52 \pm 0.32$ | $1.04 \pm 0.07$ | $1.12 \pm 0.06$ | $2.224 \pm 1.5\%$ |
| GILL | 2012-13 | 67 | $6.62 \pm 0.14$ | $0.54 \pm 0.33$ | $1.10 \pm 0.05$ | $1.08 \pm 0.04$ | $2.281 \pm 1.0\%$ |
| GILL | 2013-14 | 46 | $4.81 \pm 9.25$ | $6.52 \pm 0.77$ | $1.10 \pm 0.07$ | $1.09 \pm 0.06$ | $2.301 \pm 1.8\%$ |
| FSMI | 2011-12 | 64 | $10.35 \pm 0.16$ | $0.59 \pm 0.29$ | $1.06 \pm 0.07$ | $1.11 \pm 0.05$ | $2.257 \pm 1.4\%$ |
| FSMI | 2012-13 | 57 | $10.00 \pm 0.26$ | $0.87 \pm 0.19$ | $1.12 \pm 0.10$ | $1.07 \pm 0.06$ | $2.282 \pm 2.0\%$ |
| FSMI | 2013-14 | 54 | $10.50 \pm 0.24$ | $0.66 \pm 0.22$ | $1.12 \pm 0.09$ | $1.06 \pm 0.04$ | $2.274 \pm 1.6\%$ |




### 3.3 Magnitude variation

The signal intensity during each transit will depend on source brightness, instrument sensitivity, and atmospheric effects. This is complicated for Jupiter, as the apparent visual magnitude varies due to changes in distance from Earth. Figure 10 illustrates the importance of this effect, with predicted variation in apparent brightness following the upper bound of observations. The

5   lower set of events typically correspond to apparent transit profile widths that are significantly different than the best-case values, and are likely due to non-ideal atmospheric transmission (eg. clouds or ice crystals). There are usually several dozen "good" transits per season; subsequent analysis will focus on these events.

**Figure 10.** Peak counts from Jupiter at Gillam over three winters. Large symbols are transits with narrow widths, small symbols are noisier profiles. Solid line is variation in apparent visual magnitude of Jupiter, dashed line indicates the change in extinction due to doubling air mass ($\Delta\kappa = 0.15$).




Effects due to variation in source brightness can be removed by normalizing all measured $\bar{\mathcal{D}}$ cases to magnitude $m = 0$

$$\mathcal{D}_0 = \bar{\mathcal{D}} \times \sqrt[5]{100}^{\,m_J} \tag{32}$$

where $m_J$ is the apparent visual magnitude of Jupiter predicted by the ephemeris. The resulting distribution of normalized magnitude at Gillam (not shown) has a fairly narrow peak with a sharp higher cut-off and a long tail of lower values corresponding
5  to non-ideal viewing conditions. The 90th percentile was found to be a simple and robust estimator of peak normalized brightness, while average and standard deviation are used to estimate uncertainty in seasonal averages. Results for Gillam and Fort Smith are presented in Table 8.



**Figure 11.** Gillam transit events from Figure 10 normalized to magnitude 0 using Equation 32.

   Normalized brightness for all Gillam transits over three years are shown in Figure 11. Linear fits to the data give a slight positive slope of roughly 2% per year, but with statistical uncertainty that includes zero. This is consistent with a stable system
10  response at blue wavelengths, although variations on the order of 5% cannot be excluded.




If the linear trend were significant, this would mean the instrument was becoming slightly more sensitive over time, which seems unlikely. Closer examination of the data found that most of the variation is due to a 5% jump between 2012 and 2013 after which the signal levels remain essentially constant. The jump did not correspond to any system maintenance or modifications. A nearly identical pattern was observed at Fort Smith, further suggesting that the underlying cause was not instrumental.

In fact, this appears to be an example of atmospheric effects as discussed in §1.4.1. The apparent declination of Jupiter increased from $+5°$ in 2011 to roughly $+15°$ in 2013 and 2014. This reduced the transit zenith angle at Gillam from $\zeta = 57°$ to $44°$, and effective air mass from $X = 1.84$ to $1.39$. For a nominal blue value $\kappa = 0.17$ with zenith transmission $K = 85.5\%$ the change in declination corresponds to transmission differences of $74.9\%$ versus $80.4\%$. Adding this correction to normalized brightness reduces the linear trend to zero, although with considerable uncertainty.

## 3.4 Spectral ratio

Absolute photometric calibration with Jupiter is complicated by variability in observed brightness, and absolute spectral sensitivity is similarly challenging. Working with relative spectral response removes changes in source brightness, allowing us to focus on instrumental and atmospheric effects. In order to reduce statistical uncertainty we have normalized all channels to the average of the twin $H_\beta$ channels and summarized the results in Table 8.

Factoring out external brightness variation provides useful information about internal stability of different wavelength channels. Averages for normalized blue channels are essentially constant to within 1% year-to-year. This result provides some reassurance about relative filter stability, but cannot exclude the possibility of any change which might produce identical changes in all channels (eg. high-voltage supply drift, optical defocusing).

Red channels exhibit more variability on both short and longer time scales as shown in Figure 12. One notable feature is a clear drop after the first season, followed by two years of relative stability. This might be attributed to some wavelength dependent change in sensitivity such as photocathode aging or filter delamination. However, exactly the same pattern is observed at all four sites, suggesting a cause that is external rather than instrumental.

As noted in §1.4.1, apparent changes in wavelength ratios can also be produced by variations in source declination. Extinction at zenith will have a larger effect on shorter wavelengths, thus increasing the red:blue ratio. This effect becomes larger as zenith angle increases with largest red-to-blue ratios observed near the horizon. From 2012 to 2013 Jupiter's declination increased by roughly $10°$ and transit zenith angle decreased from $50°$ to $40°$. Assuming that observed changes in wavelength ratio are caused by this effect, a simple log-linearized regression

$$\log I_1/I_2 + \log(2.512) - x\left(\kappa_1 - \kappa_2\right) = \log D_1/D_2 \tag{33}$$

gives a slope of $\kappa_{\mathrm{red}} - \kappa_{\mathrm{blue}} \approx 0.38$ which is generally consistent with other results considered in §1.4. Since this estimate is produced by combining a large number of transits obtained during a wide range of atmospheric conditions we do not place too much weight on the precise value. The important result is that spurious trends in wavelength ratios can be modelled well enough to allow detection of real changes on the order of $5\%$.





**Figure 12.** Ratio of 630.0 nm to average of two 486.1 nm channels versus time. Large symbols correspond to good transits and small symbols to noisier events.

**Table 8.** Magnitude normalized intensity and self-normalized spectral sensitivity for Gillam and Fort Smith. Relative values calculated using all good transits during each winter.

| site | year | N | 90% | $\mathcal{D}_0$ | 471 | 480 | 486 | 486 | 495 | 558 | 625 | 630 |
|------|------|---|------|-----------------|-----|-----|-----|-----|-----|-----|-----|-----|
| GILL | 2011-12 | 73 | 530.3 | $425 \pm 143$ | 0.914 | 1.054 | 0.997 | 1.003 | 1.052 | 0.087 | 0.415 | 0.382 |
| GILL | 2012-13 | 67 | 572.7 | $474 \pm 133$ | 0.927 | 1.033 | 1.007 | 0.993 | 1.073 | 0.087 | 0.399 | 0.359 |
| GILL | 2013-14 | 46 | 582.6 | $487 \pm 141$ | 0.914 | 1.042 | 0.997 | 1.003 | 1.061 | 0.018 | 0.376 | 0.366 |
| FSMI | 2011-12 | 64 | 844.9 | $651 \pm 224$ | 0.915 | 1.063 | 1.000 | 1.000 | 1.055 | 0.071 | 0.395 | 0.379 |
| FSMI | 2012-13 | 57 | 873.2 | $732 \pm 199$ | 0.933 | 1.043 | 1.009 | 0.991 | 1.066 | 0.102 | 0.400 | 0.341 |
| FSMI | 2013-14 | 54 | 877.0 | $715 \pm 228$ | 0.907 | 1.056 | 1.003 | 0.997 | 1.049 | 0.053 | 0.387 | 0.347 |



## 3.5 Absolute sensitivity

The data count rate $\mathcal{D}$ produced by one Rayleigh per nanometer $\dot{\mathcal{R}}$ of auroral luminosity can be found by using the definition of sensitivity (Equation 30) including atmospheric losses (Equation 20) and the relationship between distributed and point sources (Equation 14)

$$\mathcal{C}_{\mathcal{D}/\dot{\mathcal{R}}} = 10^{10} \frac{\mathcal{D}}{\dot{S}_\gamma} \frac{\Omega_0}{4\pi} 2.512^{+\kappa X} \tag{34}$$

to get an expression in terms of five quantities (see also page 42 of *Wang*, 2011). Three of these terms are easily estimated, while the other two present some challenges.

The differential number flux $\dot{S}_\gamma$ of solar photons scattered from Jupiter and arriving at the top of the Earth's atmosphere is only subject to uncertainties in the solar spectrum and Jupiter's albedo, both of which are known to 1% or better. The effective air-mass $X(\zeta(t))$ depends on the apparent zenith angle which can be calculated for any arbitrary time. The effective solid angle $\Omega$ is either known *a priori* or can be estimated from transit profiles. From §3.1 the uncertainty of an unbiased estimate will be less than 1%, but systematic bias on the order of 5% is also a possibility.

The extinction coefficient spectrum $\kappa(\lambda)$ can be highly variable, can have a major effect on received signal levels, and cannot be accurately estimated from the MSP data. In the absence of other information, the best we can do is identify an upper envelope containing the brightest events and assume that they correspond to the minimum possible extinction values. This approach seems to produce intrinsic variability less than 5%, but does not address the issue of systematic bias.

Each transit could potentially provide a measured value for $\mathcal{D}$. A simple calculation of Poisson uncertainty for the entire profile would be on the order of 1% assuming good transits with peaks in excess of 2000 counts. This result may be overly optimistic given the complicated nature of many transits. An alternative approach is to examine sequences of transit profiles, focus on clusters of "bright" events in the top quartile or decile, and assume that they provide an overestimate of the intrinsic variability.This approach produces estimated uncertainties ranging from 1-5%.

Data from a single transit can be scaled by model flux density from Equation 18 to obtain an empirical estimate of the system calibration coefficient $\mathcal{C}$. An example is provided in Table 9 for the November 22, 2011 transit at Gillam using the pair of nominally identical 486 nm channels as an example. Fitting a two-dimensional Gaussian model to each channel separately produced very similar peak values: 1501.14 DN and 1501.54 DN. Appropriate model values from Table 3 can be used to predict input photon flux (neglecting atmospheric effects) and estimate a system calibration coefficient relating flux from a point source to measured data numbers.

Calculation up to this point has consisted of multiplying several quantities, each with relative uncertainty of a few percent or less. These errors are negligible in comparison to atmospheric variability. The 486.1 nm extinction factor at zenith could vary between 0.73–0.84 for fair to good visibility, and 0.64–0.78 at $\zeta = 45°$. Lower elevations and worse viewing conditions will further attenuate incoming flux. Neglecting extinction will provide a lower bound for empirical sensitivity, as reduced flux requires higher sensitivity in order to produce the same observations.

Including more events should provide some combination of additional information and increased variability. We attempt to focus on a sub-set of "high-quality" transits that presumably correspond to good atmospheric viewing conditions. Events





**Table 9.** Calibration coefficient $\mathcal{C}_{\mathcal{P}/\mathcal{D}}$ estimated at Gillam using a single transit on November 11 2011. Atmospheric effects are neglected.

| | | |
|---|---|---|
| 486.1 | [nm] | channel wavelength |
| 1501 | [DN] | peak data number |
| $5.191 \times 10^{17}$ | $[\text{photon}]/[\text{m}^2 \cdot \text{s} \cdot \text{nm}]$ | solar photon flux at 1 AU |
| $5.328 \times 10^{-10}$ | | geometric factor $A(t)$ |
| 0.455 | | jupiter albedo |
| $1.061 \times 10^9$ | $[\text{photon}]/[\text{m}^2 \cdot \text{s} \cdot \text{nm}]$ | jupiter photon flux at Earth |
| $7.067 \times 10^5$ | $[\text{photon}]/[\text{m}^2 \cdot \text{s} \cdot \text{nm} \cdot \mathcal{D}]$ | calibration coefficient $\mathcal{C}_{\mathcal{P}/\mathcal{D}}$ |
| 0.799 | | extinction at $\zeta = 45.6°$ |

are first classified according to beam widths (§3.1). Most points cluster near a common linear trend, but there are also quite a few low-brightness outliers. A robust (least absolute deviation) linear model provides a plausible fit that is insensitive to outliers. Points within a generous range around the robust fit are classified as high-quality and used for subsequent analysis, including standard least squares estimates of intercept and slope $\mathcal{C}_{\mathcal{D}/\mathcal{P}}$. Figure 13 shows classification and fitting results for the combined blue channel data. This automated process produces reasonable results for all the data considered in this study. More sophisticated algorithms for further studies could explicitly include the asymmetric nature of extinction ie. hard upper bound on theoretical maximum.

**Table 10.** Sensitivity for each channel in Data Numbers (counts) per Rayleigh per nanometer $\mathcal{C}_{\mathcal{D}/\mathcal{R}}$.

| | year | N | 471 | 480 | 486 | 486 | 495 | 558 | 623 | 630 |
|---|---|---|---|---|---|---|---|---|---|---|
| gill | 2011 | 59 | 0.2478 | 0.1816 | 0.2507 | 0.2427 | 0.2002 | 1.6296 | 1.0857 | 0.9721 |
| gill | 2012 | 60 | 0.2114 | 0.1603 | 0.1698 | 0.1702 | 0.1718 | **** | 0.8244 | 0.8764 |
| gill | 2013 | 39 | 0.1434 | 0.1280 | 0.1169 | 0.1174 | 0.1365 | 1.3462 | 0.5802 | 0.5037 |
| fsmi | 2011 | 51 | 0.0734 | 0.0707 | 0.0615 | 0.0611 | 0.0704 | 1.5203 | 0.3267 | 0.3525 |
| fsmi | 2012 | 47 | 0.1239 | 0.1182 | 0.1096 | 0.1113 | 0.1292 | 3.6000 | 0.6915 | 0.6828 |
| fsmi | 2013 | 52 | 0.1316 | 0.1222 | 0.1164 | 0.1164 | 0.1307 | 3.3278 | 0.5578 | 0.3469 |

## 4 Discussion

When auroral instruments operate unattended for long periods of time at remote locations, frequent comprehensive on-site calibration may not be feasible. If celestial objects can be identified in standard data streams then these may serve as the basis for alternative independent calibration procedures.

There is a long history of using astronomical sources to determine the alignment of auroral instruments (*Montbriand et al.*, 1965). Absolute calibration using stellar spectra appears to be a more recent development *Gladstone et al.* (2000); *Whiter et al.* (2010); *Dahlgren et al.* (2011); *Wang* (2011); *Wang et al.* (2012). Details discussions of these topics are not always found



**Figure 13.** Total counts in four blue channels (excluding 470.9 nm) as a function of predicted photon flux density. Small "+" indicate all cases, medium "x" for good beam widths, large squares for nearness to robust fit line. Flux model includes solar spectrum, illumination geometry, Jupiter albedo, and terrestrial atmospheric extinction as in Table 3.

in the primary scientific literature, but must often be extracted from conference proceedings, technical reports, and theses. Fortunately, these resources are more easily discovered with modern search engines.

Stars are essentially point sources when viewed using auroral instruments with angular resolution on the order of $1°$. They are stationary in celestial coordinates, and follow predictable paths as the Earth moves during each day and over the course of a year. Absolute flux spectra are increasingly available, although more generally for faint stars that cannot be reliably detected by most auroral devices. Even the brightest stars are only comparable to low-intensity aurora with correspondingly high statistical uncertainty. Light from extra-terrestrial sources must also travel through the Earth's atmosphere before arriving at a detector.





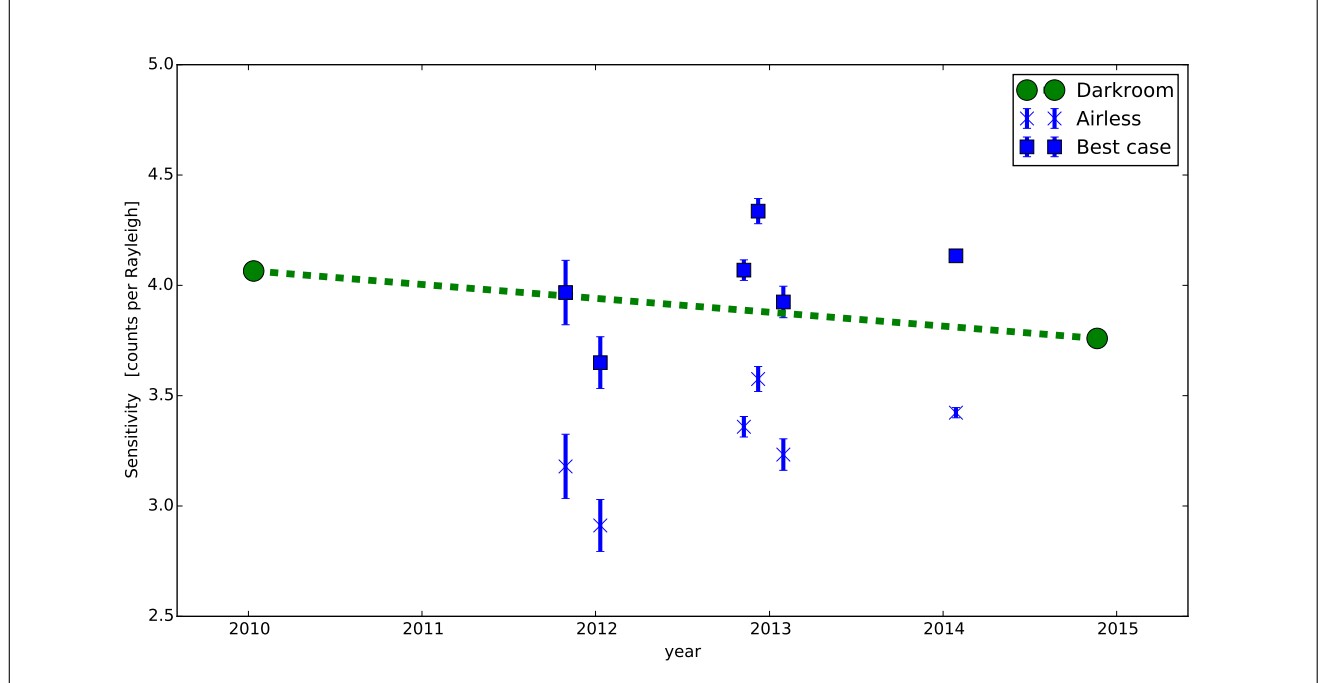

**Figure 14.** Sensitivity for the FSMI MSP 486.1 nm channels. Green circles are values obtained during darkroom calibration in 2010 and 2014, nominal linear trend of -2%/year indicated by dashed line. Blue symbols are values obtained by averaging three best values over 10-day intervals and standard deviation indicated with error bars, x's are without any atmospheric correction, squares are with "clear sky" model.

The resulting wavelength-dependent reduction in photon flux depends critically on atmospheric properties that may not be well known. Of course, auroral light is also subject to the same atmospheric effects.

Jupiter's peak radiance is greater than the brightest star, but less than the mooon, so there is no risk of saturating most auroral detectors. It is effectively point-like, has a predictable trajectory, and absolute spectral flux can be calculated from existing albedo and solar irradiance measurements. Unlike stars, planets are not fixed in celestial coordinates, meaning that transit altitude is not constant. This minor complication actually provides an opportunity to study the effects of changing zenith angle on atmospheric extinction.

## 4.1 Atmospheric effects

Atmospheric transmission is likely to be the largest source of uncertainty for high SNR applications. Reducing this uncertainty will require estimation of extinction coefficients that are appropriate for each transit. Our preliminary attempts to determine these parameters using multi-spectral MSP data were not successful, but this problem may yield to more sophisticated analysis. In principle, extinction coefficients can be found simply by measuring the apparent magnitude of a single star at a given wavelength over a range of different zenith angles. Improved precision can be achieved by combining data from multiple stars. Many auroral observatories include all-sky camera systems which can image dozens or hundreds of stars. However, the optical





response ("flat field") of these systems is also a strong function of axial angle, which for an ASI is usually directed towards the zenith. Accurate flat-fields will be essential for accurate extinction estimates. Recent work by *Duriscoe et al.* (2007); *Olmo et al.* (2008); *Román et al.* (2012) might be adapted for auroral applications.

5 It is tempting to avoid the compexity of atmospheric variation by using only a small number of "good" days to determine calibration parameters. One obvious limitation of this approach is that it cannot reliably detect short term changes in instrument response. More importantly, all auroral observations are subject to exactly the same atmospheric issues. An arc moving from the horizon to zenith will become brighter, not because of any change in precipitation, but simply due to reduction in total airmass between auroral altitudes and a ground-based observer. Atmospheric effects may be negligible when looking directly upward through clear skies, but critically important at low elevations and non-ideal viewing conditions. These effects would 10 be even more pronounced at shorter wavelengths (eg. 427.8 nm and 391.4 nm) often used in auroral studies.

## 4.2 Retrospective Calibration

Some auroral instruments only acquire data during short-term "campaigns", but many are operated in support of longer term science objectives. Not all devices are fully calibrated before being deployed and few are calibrated on a regular basis. Even when the resulting data overlap in space and time, quantitative comparison may not be possible. Astronomical observations of 15 bright sources such as Jupiter can provide a basis for retrospective cross-calibration of historical data sets.

The original CANOPUS meridan scanning photometer array (MPA) is a good example. Digital "low resolution" binned data are available starting in early 1988 and continuing until spring 2005. Some data are available for the transition period from 2005-2010, after which all refurbished instruments were operating in the same high-resolution mode. The 16 years of low-res data alone extend well beyond one solar cycle and could span more than two if merged with newer data.

20 However, certain kinds of quantitative analysis are limited by the lack of photometric calibration. Some key parameters (eg. filter band-width and channel sensitivity) were determined for each system, but the supporting documentation is very limited. Mechanical and electrical subsystems were regularly maintained and repaired, but there was no corresponding calibration schedule. Some terminal calibration procedures were carried out during the 2005-2010 transition, but by this point the instruments were often not functioning reliably. In order to confidently identify long-term geophysical trends in these data it is 25 essential to have some sense of how instrument performance changed over the same time-scales.

A preliminary survey of the CANOPUS MPA data archive has confirmed the feasibility of astronomical calibration and also identified some significant challenges. First, only the brightest few stars are visible even with optimal viewing conditions. Jupiter can be clearly identified, but at count rates much lower than obtained by the newer systems, and consequently with much greater uncertainty. Elevation steps are combined into 17 latitude bins which effectively removes the ability to determine 30 instrument tilt. More generally, it eliminates virtually all information about the optical beam-shape in that direction, including that required to estimate the effective solid angle $\Omega_0$. Finally, the decreased scan cadence of one-per-minute will slightly reduce the accuracy of azimuth estimates. Despite these limitations it should still be possible to estimate absolute sensitivity using Jupiter transits during extended intervals at both ends of the project: 1989-1993 and 1999-2005. Other bright stars or planets might be used to fill in the intervening period.



## 5 Conclusions

In this study we have demonstrated the feasibility of using Jupiter to calibrate a network of auroral meridian scanning photometers. This approach provides an estimate of instrument orientation for each transit with even marginal viewing conditions. Abrupt changes of less than $1°$ can be easily identified. If orientation is constant then it can be determined to at least $1/10°$, which exceeds most application requirements. Angular optical response (beam-shape) can be obtained from a sequence of meridian scans obtained during the transit of a point source. Statistical uncertainty may be a limiting factor even for bright stars, so the increased SNR from Jupiter is highly advantageous.

*Acknowledgements.* Funding for MSP refurbishment and ongoing operation was provided by the Canadian Space Agency under Go Canada initiative contract 13SUGOHSTO for the H STORM project. Field operations support is provided by SED systems and Athabasca University.



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
