# Peer review of "Auroral meridian scanning photometer calibration using Jupiter"

_Geoscientific Instrumentation, Methods and Data Systems, 2016_

## Referee Comment (RC1) · Anonymous Referee #1 · 21 Apr 2016

The authors report on a feasibility study for calibrating ground photometers by using the known spectrum from Jupiter in the field of view. This study includes some important advances in relating optical field measurements more regularly to a known reference and in making calibration procedures more flexible and less dependent on the laboratory facilities. The work is carefully documented and the caveats well discussed. I only have some few questions to address for clarification.

My main suggestion is to include a little summary or status report in the conclusions. That could include things like: Are both geometric and spectral calibrations based on Jupiter spectrum seen as feasible? What are the required conditions for making the calibration feasible, what do the authors see as main error sources and as the most

important area for further improvements? How much processing time does it take to perform a celestial source calibration? Would the procedure presented here be directly applicable to other optical measurements?

Minor comments:

- Figure 2: This figure is used as an example data set. Could you mark some of the main features in the plot or at least add an extra sentence or two about them for readers who are not familiar with MSP data. Useful details may be: direction of north / south (geographic or geomagnetic?), approximate luminosity range, wavelength (the title says channel 1), green/yellow feature on the right hand side edge and the bright feature at around scan number 1000. Clouds are mentioned in the text. Are there cloudy features in the sample data?

- Figure 3: Please use colors for faster distinction between the different spectra.

- Equation (16) does not explicitly include variables for solar power and planetary albedo which are explained below the equation. Please rearrange the text to match the variables in equations (16), (17) and (18).

- Figure 4: The caption has a superscript 1 which I found no further information about.

- Colors or filled/nonfilled symbols or a legend would be helpful in figures 9, 10, 11, 12 and 13 since it is challenging to tell small, medium and large symbols apart.

---

## Referee Comment (RC2) · Anonymous Referee #2 · 4 May 2016

This manuscript concerns using observations of Jupiter for calibrating groundbased meridian scanning photometers (MSP).

Using stars for geometrical calibrations of auroral imagers is well-established since a couple of decades. Using stellar spectras for absolute calibration is not as common. Relating such calibrations to laboratory calibrations with LBS is not frequently done.

The task of calibrating auroral instruments is extremely important and the authors suggest considerable improvements to existing practices. This paper should therefore be accepted after a minor revision.

Detailed minor comments and suggestions:

"1. Introduction": Well written and easy to follow but it could be improved by referring

to earlier work in the field. This is done on page 29, maybe consider moving this part here? Using stars for geometrical calibration dates back to (at least) the 1970s, several attempts has also been made over the years to use stars for absolute calibration. (Generally speaking this paper is well-referenced apart from the introduction).

page 3 line 17: "several extremely bright lines and bands from atomic oxygen and molecular nitrogen" "extremely bright" is maybe exaggerating a bit. Relatively speaking it is correct, but not even the brightest line of atomic oxygen is "extremely bright", not to mention first negative at 427.8 nm Please consider rephrasing this.

page 5, line 4: What does "luminosity" stand for here? Please clarify. Total energy emitted by the object?

page 5-6 1.2.1 Geometric: Please add suitable references to this section.

page 7, line 9 "Photometry" Please consider using "Radiometry" instead. Photometry is easily confused with photometric units, which are irrelevant here.

page 7 Eq(8): Something is wrong here. Radiance has units watts per (squaremeter steradian). Either sr is missing or the authors intended something else. The symbol L is commonly used for radiance. Please correct or clarify. See also below. How is radiance of a point-source defined?

page 7, equation (10) The column emission rate is $4\pi L = \int_0^\infty \ldots$ (see Hunten 1956)

page 7, line 19: "has units of radiance" is the apparent radiance. (See Baker& Romick 1976)

This section (1.2.3) could maybe be clarified by starting with the basic quantity radiance (L) of an extended source (aurora), then discuss the Rayleigh and proceed to irradiance (E) at the detector. Then treat the case of a point source and $1/r^2$.

page8, lines 18–19: "Only the brightest stars can produce count rates comparable to background contributions such as airglow. " Incorrect! Typically hundreds of stars per

image are normally used for geometrical calibration of images from imagers equipped with narrow-band interference filters.

page 9 table 2: $[sm^2\ nm]^{-1}$ is centered above [J] and [#] looks strange.

page 10 lines 14–15: "...is still a hundred times brighter than the brightest aurora."

IBC-IV aurora (1 MR at 557.7 nm) is often compared to the luminous intensity of the full moon (0.1 Lux for a human observer) . This doesn't make sense with "a hundred times brighter"

page 17 Eq. (30): No reason to use the inverse of Eq(29).

page 28, Eq(34): This equation is central to the paper and should be discussed in greater detail.

page 29 line 12 – page 31 line 7: Please consider moving (parts of) this text to the introduction.

page 32 lines 6–8 "An arc moving from the horizon to zenith will become brighter, not because of any change in precipitation, but simply due to reduction in total airmass between auroral altitudes and a ground-based observer." Correct, but please also consider number of photons integrated when looking along the magnetic field-line instead of across it. This is the main cause of the intensification.

page 33 conclusions: Maybe summarize a bit better, and/or include a small table of the most important results. Future outlook?

Table 8: clarify units.

Figure 1: reproduces badly and lacks site mnemonics (RANK, GILL, etc.)

Figure 2: Keogram empty in printout. Looks good in PDF.

Figure 8. x and y labels in the figure could be improved.

---

## Author Comment (AC1) · 11 Jul 2016

We sincerely appreciate the detailed comments by the reviewers which have resulted in substantial improvements to the manuscript. Specific responses to each point are included below. Changes to the manuscript are indicated in a PDF generated with "latexdiff".

Reviewer #1 ==========================================

The authors report on a feasibility study for calibrating ground photometers by using the known spectrum from Jupiter in the field of view. This study includes some important advances in relating optical field measurements more regularly to a known reference and in making calibration procedures more flexible and less dependent on the laboratory facilities. The work is carefully documented and the

caveats well discussed. I only have some few questions to address for clarification.
==========================================
* * *
My main suggestion is to include a little summary or status report in the conclusions. That could include things like: Are both geometric and spectral calibrations based on Jupiter spectrum seen as feasible? What are the required conditions for making the calibration feasible, what do the authors see as main error sources and as the most important area for further improvements? How much processing time does it take to perform a celestial source calibration? Would the procedure presented here be directly applicable to other optical measurements?

—Yes, several paragraphs added to conclusion.

Minor comments:

Figure 2: This figure is used as an example data set. Could you mark some of the main features in the plot or at least add an extra sentence or two about them for readers who are not familiar with MSP data. Useful details may be: direction of north / south (geographic or geomagnetic?), approximate luminosity range, wavelength (the title says channel 1), green/yellow feature on the right hand side edge and the bright feature at around scan number 1000. Clouds are mentioned in the text. Are there cloudy features in the sample data?

—We have selected an example with more complex features eg. moonlight; annotations have been added to indicate these features.

Figure 3: Please use colors for faster distinction between the different spectra.

—Done.

Equation (16) does not explicitly include variables for solar power and planetary albedo which are explained below the equation. Please rearrange the text to match the vari-

ables in equations (16), (17) and (18).

—Fixed.

Figure 4: The caption has a superscript 1 which I found no further information about.

—Fixed.

Colors or filled/nonfilled symbols or a legend would be helpful in figures 9, 10, 11, 12 and 13 since it is challenging to tell small, medium and large symbols apart.

—Legends added to several figures.

Please also note the supplement to this comment:
http://www.geosci-instrum-method-data-syst-discuss.net/gi-2016-5/gi-2016-5-AC1-supplement.pdf

**Supplement:**

**Auroral meridian scanning photometer calibration using Jupiter**

Brian J. Jackel[1], Craig Unick[1], Fokke Creutzberg[2], Greg Baker[1], Eric Davis[1], Eric F. Donovan[1], Martin Connors[3], Cody Wilson[1], Jarrett Little[1], M. Greffen[1], and Neil McGuffin[1]

[1]University of Calgary, Alberta, Canada
[2]Natural Resources Canada Geomagnetism Laboratory
[3]Athabasca University, Alberta, Canada

*Correspondence to:* Brian J. Jackel
brian.jackel@ucalgary.ca

**Abstract.** Observations of astronomical sources  provide information that can significantly enhance the utility of auroral data for scientific studies.  This report presents results obtained by using Jupiter for field cross-calibration of 4 multi-spectral auroral meridian scanning photometers during 2011-15 northern hemisphere winters. Seasonal average optical field-of-view and local orientation estimates are obtained with uncertainties of $0.01°$ and $0.1°$ respectively. Estimates of absolute  sensitivity are repeatable to roughly 5% from one month to the next, while the relative response between different wavelength channels is stable to better than 1%. Astronomical field calibrations and darkroom calibration differences are on the order of 10%. Atmospheric variability is the primary source of uncertainty; this may be reduced with complementary data from co-located instruments.

**1 Introduction**

Interactions between the solar wind and the terrestrial magnetic field produce a complex and dynamic geospace environment. Ionospheric phenomena such as the aurora are connected to magnetospheric processes by mass and energy transport along magnetic field lines. Consequently, auroral observations provide information that can be used for remote sensing of distant plasma structure and dynamics. A single ground-based instrument can only view a small part of the global system. , so a combination of instruments at different locations (eg. Figure 1 and Table 1)  multiple data sets requires accurate information about device characteristics such as timing, orientation, and absolute spectral sensitivity.

Comprehensive calibration requires specialized equipment and skilled personnel that are typically available only at centrally located research facilities. With sufficient resources it is possible, at least in principle, to determine all device parameters that are required to convert raw instrument data numbers to physically useful quantities. Practical limitations can result in random or systematic uncertainties which may impede quantitative scientific analysis. This is particularly relevant for large networks of nominally identical instruments, where ongoing calibration of each device may be extremely challenging.

Even assuming ideal calibration at a central facility, many auroral instruments must be operated at remote field sites. Transfer between these locations requires a sequence of packing, shipping, and re-assembly that is time-consuming, costly, and may

[Figure]

**Figure 1.** Canadian meridian scanning photometer site locations (details in Table 1). Fan shapes indicate $4°$ optical beam width for altitudes of 110 and 220 km at elevations of $10°$ above the horizon.  Dashed contours indicate magnetic dipole latitude (IGRF 2015).

**Table 1.** Canadian meridian scanning photometer site information. Geographic latitude, longitude, and altitude are in degrees North, degrees East, and metres above mean sea level (WGS-84). L-shell and magnetic declination  from the IGRF model.

| | Geographic | | | L-shell | | Declination | | |
|---|---|---|---|---|---|---|---|---|
| | Lat | Lon | Alt | 1988 | 2013 | 1988 | 2013 | |
| RANK | 62.82 | 267.89 | 32 | 11.20 | 10.64 | -7.1 | -7.7 | Rankin Inlet, Nunavut |
| GILL | 56.35 | 265.29 | 99 | 6.04 | 5.83 | 2.6 | -0.5 | Gillam, Manitoba |
| PINA | 50.20 | 263.96 | 262 | 3.95 | 3.84 | 5.5 | 2.3 | Pinawa, Manitoba |
| FSMI | 60.02 | 248.05 | 205 | 6.65 | 6.58 | 24.3 | 15.8 | Fort Smith, NWT |
| ATHA | 54.70 | 246.70 | 533 | 4.50 | 4.45 | 21.1 | 15.3 | Athabasca, Alberta |

unintentionally alter instrument response. Furthermore, intermittent calibration cannot distinguish between a gradual drift or sudden changes.

Extra-terrestrial sources, such as planets or stars,  are often used for calibration of spatially resolved optical or radio frequency data. Instrument orientation can be determined from objects whose positions are well known, while source inten-
5  sity can be used to verify instrument sensitivity. Astronomical sources are often detectable in existing auroral data streams, allowing for ongoing monitoring of system response and the possibility of retrospective re-analysis of older data sets. Practical application may be restricted by instrumental limitations and complications including man-made interference, clouds, aurora and other geophysical processes.

There is a long history of using astronomical sources to determine the alignment of auroral instruments (**????**). Absolute
10  calibration using stellar spectra appears to be a more recent development (**??????**). Detailed discussions of these topics are not always provided in the primary scientific literature, but must often be extracted from conference proceedings, technical reports, and theses.

The focus of this paper is on the field calibration of a network of four auroral photometers using Jupiter as a standard reference. Some key features of optical aurorae are provided in Section 1.1, §1.2 introduces key calibration concepts and
15  results, essential astronomical topics are presented in §1.3, and atmospheric effects are briefly reviewed in §1.4. An overview of instrument details is given in §2, data analysis and results are in §3, discussion in §4, followed by a summary and conclusions in §5.

**1.1 Optical Aurora**

In regions of geospace where magnetic field lines can be traced to the Earth, some charged particles may travel down to
20  altitudes where neutral densities are no longer negligible. Collisions with atmospheric atoms or molecules may transfer energy which can be re-emitted as photons. Spectral, spatial, and temporal features of the optical aurora contain information about geospace plasma properties, allowing for remote sensing of magnetospheric topology and dynamics.

Auroral spectra are dominated by several  relatively bright lines and bands from atomic oxygen and molecular nitrogen, with many other less intense features ranging from extreme ultra-violet through to far infra-red. The intensity of auroral emission at different wavelengths depends on precipitation energy and atmospheric composition, as more energetic particles are able to penetrate to lower altitudes where constituents may be more or less abundant. Consequently, observations at multiple wavelengths can be combined to infer characteristics of the precipitating particles (**??**). These multi-spectral measurements can be challenging due to the wide dynamic range between very bright 558 nm "green-line" (1-100 kiloRayleigh) emissions and extremely faint 486 nm "proton aurora" ($<$ 100 Rayleighs).

Optical aurora typically occur within "auroral ovals", roughly centered around each geomagnetic pole, extending hundreds of kilometres in latitude and thousands of kilometres in longitude (**?**). Luminosity can be highly dynamic over a wide range of spatial scales, but quiet-time structures generally exhibit a narrow latitudinal extent (10's to 100's of km) and relatively less longitudinal variation over 100's or 1000's of km (**?**). This spatial anisotropy is one motivation for using a meridian scanning photometer (MSP, see §2) to measure auroral luminosity as a sequence of latitude profiles (keogram). As shown in Figure 2, this data can also be used to identify other non-auroral features such as clouds and stars.

**1.2 Instrument Calibration**

Optical designs can be  modeled very precisely with modern software tools, but instrument calibration provides essential information about the actual performance. System response is not necessarily constant, but can change either gradually (eg. filter bandpass drift, decreased detector sensitivity) or abruptly (eg. damage during shipping). Such problems could be identified with calibration of instruments in the field. This process must be completely automatic, as many remote sites do not have full-time technical staff. It should be repeated frequently in order to identify abrupt changes in system response, but without interrupting or degrading normal data acquisition. A regular schedule of measurements with portable low-brightness sources (LBS) might satisfy some of these requirements, but would involve a substantial allocation of resources for repeated site visits.

In this report we examine some of the strengths and limitations of astronomical calibration for auroral instruments. We focus on issues related to field cross-calibration of MSPs which have been used extensively for auroral research (see §2 for details). However, many of these topics can also be applied more generally to other instruments used to study the optical aurora, such as all-sky imagers (ASIs).

A single ground-based instrument may measure photons with wavelengths $\lambda$ arriving from angular locations $\theta, \phi$.  The distribution of incident light $I$ is convolved with the instrument response function $f$ to product a measurement $M$ with error $M_\epsilon$

$$M(\theta, \phi, \lambda) = f(\theta, \phi, \lambda) \star I(\theta', \phi', \lambda') + M_\epsilon(\theta, \phi, \lambda) \tag{1}$$

For an ideal device $f$ would be a delta function and $M = I$, but any real measurement will have limited resolution. The goal of calibration or characterization is to determine the instrument response function $f$ in order to better understand the "true" source properties.

[Figure]

**Figure 2.** Keogram from meridian scanning photometer operating at Gillam during the night of December  20 2012 from  0000 UT  to dawn at 1320. Local midnight is approximately 0600 (scan number 720).  Data counts have been clipped and  log-scaled in  to  display Jupiter, stars, aurora, full moon, and  dawn.

[revised manuscript text omitted]

**1.2.3 Radiometry**

 At a distance $R$ from an isotropic point source with total power output  $P_0$  the irradiance (intensity) $S$  will be

$$S = \frac{P_0}{4\pi R^2} \qquad \text{watt} \cdot \text{meter}^{-2} \tag{8}$$

so that an observer at some distance $r$ will intercept an amount of power

$$P_\delta = S\, A_{\text{eff}} \tag{9}$$

proportional to the effective receiver surface area $A_{\text{eff}}$ .

Power from an extended source can be expressed in terms of a volume emission rate $\rho(r,\theta,\phi)$ integrated over the entire source region weighted by the receiver angular sensitivity $G(\theta,\phi)$

$$P_V = \oiint d\Omega\, \frac{L\,G}{4\pi} \qquad 4\pi L \equiv \int_0^\infty dr\, \rho(r) \tag{10}$$

where the radial integral $L$ has units of radiance (watt $\cdot$ meter$^{-2}$ $\cdot$ sr$^{-1}$) and is often referred to as the "column emission rate". For a uniform source radiance the total received power

$$P_V = \oiint d\Omega\, \frac{L(\theta,\phi)}{4\pi} A_{\text{eff}} \hat{G}(\theta,\phi) \approx L\ A_{\text{eff}}\, \Omega_0 \tag{11}$$

depends on the product of the effective area and the effective solid angle.  For any signal detected from some point source  there will be an equivalent volume emission which would produce the same observed power. For a uniform emission region the  relationship

$$P_\delta = P_V \qquad \rightarrow \qquad L = \frac{S}{\Omega_0} \tag{12}$$

depends only on the effective solid angle.

Auroral  intensity $\mathcal{I}$ is customarily expressed in units of Rayleighs (**????**) which is related to photon radiance $L_\gamma$ via

$$4\pi L_\gamma(\lambda) \equiv \mathcal{I}(\lambda) \qquad 10^{10}\ \text{photon} \cdot \text{s}^{-1} \cdot \text{m}^{-2} \tag{13}$$

where the subscript $E$ indicates energy flux and $\gamma$ is photon number flux. For narrow-band channels

$$\mathcal{I}(\lambda) = \int \dot{\mathcal{I}}(\lambda) \approx \dot{\mathcal{I}}\, \Delta\lambda = 4\pi \frac{\dot{S}_E}{\Omega_0} \frac{\lambda}{h\,c} \Delta\lambda \tag{14}$$

converting differential radiant spectral density $\dot{S}$ to equivalent Rayleighs per nanometer $\dot{\mathcal{I}}$ requires only the effective solid angle, which can also be estimated from observations of a point source. Working with Rayleighs requires some additional knowledge in the form of the effective bandwidth $\Delta\lambda$. As this is also true for darkroom LBS calibration, we focus here on relating $\dot{I}$ in Rayleighs per nanometer to $\dot{S}$ in Watts per metre-squared per nanometer.

**1.3 Astronomical sources**

Extra terrestrial objects have many properties which are required for accurate calibration. Locations in the celestial sphere are known to arc-second resolution or better, which is  sufficient for determining orientation and geometric response of most auroral instruments. Absolute spectral irradiance profiles are available for many sources, providing opportunities for  radiometric calibration of narrow-band instruments. Total visible intensity of most sources is essentially constant, allowing for long term monitoring of system performance. A single object can be viewed simultaneously by multiple instruments at nearby sites, facilitating quantitative inter-comparisons.

Most astronomical objects are effectively point sources,  and under good viewing conditions modern all-sky imagers can resolve hundreds of stars with a relatively short exposure time. Ironically, the presence of bright aurora or airglow can be a major source of error in radiometric calibration. For the MSP considered here, the total light from Vega passing through through a 3 nm filter is approximately 200 Rayleighs, which is comparable to typical red-line airglow emissions. Even on a moonless night, continuum emissions can be on the order of 10 R/nm, equivalent to stars of magnitude 2 as observed by our MSP. Note that there are only 50 stars of magnitude 2 or brighter, and fewer than half of them are visible from the northern auroral zone at any given time.

Celestial source brightness spans a wide range and is usually expressed in terms of logarithmic magnitude $m$

$$I = \sqrt[5]{100}^m \approx 2.512^m \tag{15}$$

so that the relative intensity of two sources can be determined from the difference of their magnitudes. Absolute flux distributions as a function of wavelength are available for most of the brightest stars, including Vega (**?**), Sirius (**?**), and Arcturus (**??**). Other catalogs contain many other stars (**?????**), but the majority may be too dim for reliable observation by typical auroral instruments.

**Table 2.** Selected astronomical source irradiance at Earth. Energy flux is Joules per $[s \cdot m^2 \cdot nm]$ and number flux is photons per $[s \cdot m^2 \cdot nm]$ Rayleighs are for a viewing solid angle of $\Omega = 0.002$ steradians ($2.9°$ of arc).

|         | [nm] | [$J$]    | [#]      | [R / nm] |
|---------|------|----------|----------|----------|
| jupiter | 486  | 4.78e-10 | 1.17e+09 | 735      |
| jupiter | 556  | 5.45e-10 | 1.53e+09 | 958      |
| sirius  | 556  | 1.35e-10 | 3.78e+08 | 237      |
| vega    | 556  | 3.44e-11 | 9.63e+07 | 60.5     |
| moon    | 556  | 4.63e-06 | 1.3e+13  | 8.14e+06 |
| sun     | 556  | 1.81     | 5.07e+18 | 3.18e+12 |

Conversely, the sun is so bright that direct observation will saturate detectors designed for relatively faint aurora. **?** provide an absolutely calibrated distribution of flux versus wavelength at 1 AU with sub-nanometer spectral resolution. For a nominal

instrument solid angle of 2 milli-steradians (3° of arc) the apparent solar brightness at 556 nm is roughly 3 teraRayleighs per nanometer (Table 2). Daytime operations are only possible for systems that respond to an extremely narrow range of wavelengths (**?**).

[Figure]

**Figure 3.** Spectra of solar irradiance ( green shaded curve)  from **?** and Jupiter albedo (blue line) from **?**. Inset  displays the same quantities  for the range of wavelengths associated with most visible aurora.

[revised manuscript text omitted]

detection of narrow emission lines.  An effective passband

$$\Delta\lambda_j = \int d\lambda\, \hat{T}_j(\lambda) \tag{25}$$

is the relevant quantity for broad-band calibration sources ie. converting from Rayleighs per nanometer to Rayleighs. These data suggest typical passband and transmission variations on the order of 5% between different sets of filters.

**Table 5.** Characteristics of three sets of nominally identical narrow band filters. Passband is integral of transmission profile, 90% bandwidth is the range between 5% and 95% points of the cumulative transmission.

| [nm] | passband [nm] | | | peak transmission [%] | | | 90% bandwidth [nm] | | |
|---|---|---|---|---|---|---|---|---|---|
| 470.9 | 2.483 | 2.362 | 2.355 | 82.94 | 79.36 | 78.28 | 3.60 | 3.40 | 3.50 |
| 480 | 2.592 | 2.418 | 2.661 | 85.59 | 78.60 | 87.74 | 3.10 | 3.20 | 3.20 |
| 486.1 | 2.605 | 2.587 | 2.615 | 88.26 | 85.95 | 87.57 | 2.90 | 3.00 | 3.00 |
| 486.1 | 2.572 | 2.222 | 2.509 | 84.71 | 74.21 | 83.44 | 3.10 | 3.00 | 3.10 |
| 495 | 2.607 | 2.525 | 2.584 | 88.40 | 85.74 | 87.27 | 3.30 | 3.40 | 3.50 |
| 557.7 | 1.788 | 1.728 | 1.920 | 82.93 | 78.93 | 88.53 | 4.60 | 4.90 | 4.30 |
| 625 | 1.624 | 1.632 | 1.588 | 84.46 | 87.37 | 86.09 | 4.20 | 3.20 | 2.80 |
| 630 | 1.597 | 1.590 | 1.558 | 86.67 | 84.83 | 83.81 | 2.40 | 2.60 | 2.40 |

Light which passes through the filters is detected by a photo-multiplier tube (PMT) with photocathode quantum efficiency ranging from 20% at 400 nm to 2% at 750 nm; this response was selected to maximize response for the faint $H_\beta$ emissions. A dynode chain amplifies each electron to produce a cascade which triggers a pulse-counting circuit. The high-voltage power supply required for this process is quite stable over short intervals under ideal conditions, but may change during extended field operations. Photocathode aging and high-voltage drift are likely to be the primary  causes of any long-term reduction in system sensitivity.

PMTs dead-time produces a non-linear response at high count-rates. This pulse pile-up effect can be largely removed if the time resolution $\tau$ of the system is known and is not significantly longer than the signal count interval. For the PMTs used in this study nonlinearity only becomes important for count rates greater than $10^5$ photons per second. These rates can be produced by very bright aurora but are not a problem for any astronomical sources except the Sun and Moon.

Meridian scans are achieved with a $45°$ tilted mirror and a stepping motor. Many MSPs rotate the mirror at a fixed rate in order to produce data from evenly spaced elevations. Both the original and refurbished systems considered here instead utilize a sequence of variable steps chosen to produce nearly constant exposure times as a function of linear distance at auroral altitudes. This detail is relevant to this study because Jupiter transit profiles will be measured with different resolution depending on transit elevation. The effects are expected to be small, but must be kept in mind when considering multi-year variability.

**2.1 System sensitivity**

The relationship between incident photon flux $\mathcal{P}(\lambda)$ and measured channel count rate $\mathcal{D}_k$

$$\mathcal{D} = A_{\text{eff}}\, M_x\, \Delta t \int d\lambda\, \mathcal{P}(\lambda)\, T_k(\lambda)\, Q(\lambda) \tag{26}$$

depends on the effective aperture allowing photons into the system ($A_{\text{eff}}$), channel multiplexing efficiency ($M_k$), filter transmission ($T_k$), measurement interval ($\Delta t$), and the detector efficiency $Q(\lambda)$.

For wide-band input through narrow-band filters the process can be written in terms of filter peak transmission $T_k$ and bandwidth $\Delta\lambda_k$

$$\mathcal{D}(\lambda_i) \approx \mathcal{P}(\lambda_k)\, A_{\text{eff}}\, M_k\, \Delta\lambda_k\, T(\lambda_k)\, \Delta t\, Q(\lambda_i) \tag{27}$$

 from which we can isolate a coefficient of response $_k\mathcal{C}_{\mathcal{D}/\mathcal{P}}$ for each channel

$$_k\mathcal{C}_{\mathcal{D}/\mathcal{P}} = \frac{\mathcal{D}(\lambda_k)}{\mathcal{P}(\lambda_k)}$$
$$= A_{\text{eff}}\, M_x\, \Delta\lambda_k\, T(\lambda_k)\, \Delta t\, Q(\lambda_k) \tag{28}$$

in terms of measured $\mathcal{D}$ and predicted $\mathcal{P}$ for each filter wavelength.  In principle this equation could be used to calculate coefficients in terms of  fundamental properties of each instrument. In practice, calibration coefficients are often estimated empirically by measuring sources with known brightness. For auroral applications the goal is to determine the differential sensitivity $\mathcal{C}_{\mathcal{D}/\mathcal{R}}$ relating data numbers to Rayleighs per nanometer.

[revised manuscript text omitted]

Fortunately, it is possible to accurately determine instrument orientation from transit observations. Starting with site locations obtained using GPS, observed transit times were used to calculate the actual elevation and azimuth of Jupiter for each night. These were interpreted in terms of two device angles. First, azimuth offset was attributed to horizontal orientation of a level instrument. Second, the difference between nominal mirror elevation and actual target elevation was attributed to instrument 5 "tilt" from level.

Results for in Figure 9.  several seasons of azimuth estimates at Gillam (not shown) are extremely stable over time, with jitter $< 1°$ and no apparent drift. Tilt estimates  are shown in in Figure 9. The first two seasons are generally stable, although there appears to be a small jump in early November. Examination of results from the other three sites (not shown) finds a similar feature at Fort Smith (FSMI), a smaller shift 10 at Pinawa (PINA), and no obvious change at Athabasca (ATHA). These results are consistent with "frost heave" occurring in early winter as moisture in the soil freezes. The lack of this effect at ATHA may be be due to better foundations for the instrument platform. The large change in tilt at Gillam during summer 2013 occurred around the same time as a maintenance trip. This shift could not have been detected in real-time due to the lack of Jupiter transit data during limited observing hours during summertime operations. Fortunately, once the problem has been identified, it is relatively straightforward to make the 15 necessary corrections to scientific data products.

[revised manuscript text omitted]
  extended luminosity. This can be related to the differential irradiance of an ideal point source using Equation 14. Losses due to atmospheric effects can be

[Figure]

**Figure 12.** Ratio of 630.0 nm to average of two 486.1 nm channels versus time. Large symbols correspond to good transits and small symbols to noisier events.

modelled with Equation 20. The combination of these three equations

$$\mathcal{C}_{\mathcal{D}/\dot{\mathcal{R}}} = 10^{10} \frac{\mathcal{D}}{\dot{S}_\gamma} \frac{\Omega_0}{4\pi} 2.512^{+\kappa X} \tag{32}$$

 gives an expression for calibration coefficients in terms of five physical quantities (see also page 42 of **?**). Three of these terms are easily estimated, while the other two present some challenges.

5    The differential number flux $\dot{S}_\gamma$ of solar photons scattered from Jupiter and arriving at the top of the Earth's atmosphere is only subject to uncertainties in the solar spectrum and Jupiter's albedo, both of which are known to 1% or better. The effective air-mass $X(\zeta(t))$ depends on the apparent zenith angle which can be calculated for any arbitrary time. The effective solid angle $\Omega$ is either known *a priori* or can be estimated from transit profiles.  ; from §3.1 the uncertainty of an unbiased estimate will be less than 1%, but systematic bias on the order of 5% is also a possibility.

**Table 8.** Magnitude normalized intensity and self-normalized spectral sensitivity for Gillam and Fort Smith. Column 4 is the 90th percentile of intensity. Column 4 is the source normalized brightness (Equation 30). Remaining columns are channel brightness normalized to average of two 486 nm observations.

[revised manuscript text omitted]
. A constant emission feature moving from the horizon to zenith will appear brighter even after accounting for viewing geometry (i.e. Van Rhijn correction) simply due to the reduction in total airmass between auroral altitudes and a ground-based observer. Atmospheric effects may be negligible when looking directly upward through clear skies, but critically important at low elevations and non-ideal viewing conditions. These effects would be even more pronounced at shorter wavelengths (eg. 427.8 nm and 391.4 nm) often used in auroral studies.

**4.2 Retrospective Calibration**

Some auroral instruments only acquire data during short-term "campaigns", but many are operated in support of longer term science objectives. Not all devices are fully calibrated before being deployed and few are calibrated on a regular basis. Even when the resulting data overlap in space and time, quantitative comparison may not be possible. Astronomical observations of bright sources such as Jupiter can provide a basis for retrospective cross-calibration of historical data sets.

The original CANOPUS meridian scanning photometer array (MPA) is a good example. Digital "low resolution" binned data are available starting in early 1988 and continuing until spring 2005. Some higher resolution data are available for the transition period from 2005-2010, after which all refurbished instruments were operating in the same high-resolution mode. The 16 years of low-res data alone extend well beyond one solar cycle and could span more than two if merged with newer data.

However, certain kinds of quantitative analysis are limited by the lack of radiometric calibration. Some key parameters (eg. filter band-width and channel sensitivity) were determined for each system, but the supporting documentation is very limited. Mechanical and electrical subsystems were regularly maintained and repaired, but there was no corresponding re-calibration schedule. Some terminal calibration procedures were carried out during the 2005-2010 transition, but by this point the instruments were often not functioning reliably. In order to confidently identify long-term geophysical trends in these data it is essential to have some sense of how instrument performance changed over the same time-scales.

A preliminary survey of the CANOPUS MPA data archive has confirmed the feasibility of astronomical calibration and also identified some significant challenges. First, only the brightest few stars are visible even with optimal viewing conditions. Jupiter can be clearly identified, but at count rates much lower than obtained by the newer systems, and consequently with much greater uncertainty. Elevation steps are combined into 17 latitude bins which effectively removes the ability to determine instrument tilt. More generally, it eliminates virtually all information about the optical beam-shape in that direction, including that required to confidently estimate the effective solid angle $\Omega_0$. Finally, the decreased scan cadence of one-per-minute will slightly reduce the accuracy of azimuth estimates. Despite these limitations it should still be possible to estimate absolute

sensitivity using Jupiter transits during extended intervals at both ends of the project: 1989-1993 and 1999-2005. Other bright stars or planets might be used to fill in the intervening period.

**5 Conclusions**

In this study we have demonstrated the feasibility of using Jupiter to calibrate a network of auroral meridian scanning photometers.  During times when Jupiter is visible in the night sky it can be easily distinguished from other astronomical sources. Statistical uncertainty may be a limiting factor even for bright stars, so the increased signal from Jupiter is highly advantageous. Addition precision can be achieved by combining results from multiple days with good viewing conditions.

For geometric calibration, this approach provides an estimate of instrument orientation for each transit with even marginal viewing conditions.  Changes of less than $1°$ between successive transits can be easily identified.  Absolute orientation can be determined to at least $1/10°$, which exceeds most application requirements. Angular optical response (beam-shape) can be  estimated to roughly 1% precision by combining several dozen transits.

Relative spectral calibrations (ratios of different channels) can also be obtained with precisions on the order of 1% during a single field season. Absolute radiometric calibration for individual channels is significantly less precise. This is due primarily to the difficulty of obtaining and identifying perfectly "clean" transits. Even results from apparently ideal transits can differ by 5-20%, likely due to uncertainties in the true atmospheric extinction parameters.

The merits of Jupiter as a calibration source also apply to other types of auroral instruments. Utility of stellar calibration for all-sky imagers has been demonstrated (**???**) and these methods would be even more effective with a brighter source. Given the complexities of absolute calibration, it might be helpful if observations were presented in some standard format, eg. data numbers normalized to source magnitude $\mathcal{D}_0$ as defined in Equation 30. This, along with estimates of solid angle $\Omega_0$ and bandwidth $\Delta\lambda$, would greatly facilitate the inter-comparison of different data products, which would be beneficial for both instrument operators and end-users of scientific data products.

In principle, astronomical calibration could be extracted from almost any auroral data set. In practice, this process is typically applied on a case-by-case basis and requires a considerable amount of human intervention and instrument specific knowledge. The essential next step is to develop automated software tools which can be applied more broadly. This will significantly increase the utility of optical auroral observations for quantitative scientific analysis.

*Acknowledgements.* Funding for  MSP refurbishment and ongoing operation was provided by the Canadian Space Agency under Go Canada initiative contract 13SUGOHSTO for the H STORM project. Field operations support is provided by SED systems and Athabasca University.

---

## Author Comment (AC2) · 11 Jul 2016

We sincerely appreciate the detailed comments by the reviewers which have resulted in substantial improvements to the manuscript. Specific responses to each point are included below.

Reviewer#2 ====================== This manuscript concerns using observations of Jupiter for calibrating groundbased meridian scanning photometers (MSP). Using stars for geometrical calibrations of auroral imagers is well-established since a couple of decades. Using stellar spectras for absolute calibration is not as common. Relating such calibrations to laboratory calibrations with LBS is not frequently done. The task of calibrating auroral instruments is extremely important and the authors suggest considerable improvements to existing practices. This paper should therefore be

accepted after a minor revision. ========================

Detailed minor comments and suggestions:

"1. Introduction": Well written and easy to follow but it could be improved by referring to earlier work in the field. This is done on page 29, maybe consider moving this part here? Using stars for geometrical calibration dates back to (at least) the 1970s, several attempts has also been made over the years to use stars for absolute calibration. (Generally speaking this paper is well-referenced apart from the introduction).

—Yes. The paragraph with key references has been moved from the discussion to the introduction.

page 3 line 17: "several extremely bright lines and bands from atomic oxygen and molecular nitrogen" "extremely bright" is maybe exaggerating a bit. Relatively speaking it is correct, but not even the brightest line of atomic oxygen is "extremely bright", not to mention first negative at 427.8 nm Please consider rephrasing this.

—Good point. New text: " Auroral spectra are dominated by several relatively bright lines and bands from atomic oxygen and molecular nitrogen, with many other less intense features ranging from extreme ultra-violet through to far infra-red. "

Page 5, line 4: What does "luminosity" stand for here? Please clarify. Total energy emitted by the object?

—"Spectral radiance" was intended, but now we simply say "distribution of incident light".

page 5-6 1.2.1 Geometric: Please add suitable references to this section.

—Added references to seminal work by Stormer, Chapman etc. on page 3.

page 7, line 9 "Photometry" Please consider using "Radiometry" instead. Photometry is easily confused with photometric units, which are irrelevant here.

—Good point. Several other instances of "photometry" and "photometric" have also been replaced with radiometry and radiometric.

page 7 Eq(8): Something is wrong here. Radiance has units watts per (squaremeter steradian). Either sr is missing or the authors intended something else. The symbol L is commonly used for radiance. Please correct or clarify. See also below. How is radiance of a point-source defined?

—Yes, it should have been "irradiance". We have reviewed all other instances of "radiance" and "irradiance" and believe that they are used appropriately.

Page 7, equation (10) The column emission rate is 4L =R 10 : : : (see Hunten 1956)

—Yes, fixed.

page 7, line 19: "has units of radiance" is the apparent radiance. (See Baker& Romick 1976) This section (1.2.3) could maybe be clarified by starting with the basic quantity radiance (L) of an extended source (aurora), then discuss the Rayleigh and proceed to irradiance (E) at the detector. Then treat the case of a point source and 1=r2.

—When writing the paper we tried both approaches, and decided that starting with a point source was better suited for this case. Could the reviewer suggest a reference to the other approach?

Page8, lines 18ÂŰ19: "Only the brightest stars can produce count rates comparable to background contributions such as airglow. " Incorrect! Typically hundreds of stars per image are normally used for geometrical calibration of images from imagers equipped with narrow-band interference filters.

—Yes, under very good viewing conditions a modern narrow-band ASI can resolve many hundreds of stars. However, for red-line observations at 630nm with a 3nm pass-band, a few hundred Rayleighs of airglow can make it much more difficult to resolve more than a few dozen of the brightest stars. New text: "" Most astronomical objects are effectively point sources, and under good viewing conditions modern all-sky im-

Interactive
comment

agers can resolve hundreds of stars with a relatively short exposure time. Ironically, the presence of bright aurora or airglow can be a major source of error in radiometric calibration. For the MSP considered here, the total light from Vega passing through through a 3 nm filter is approximately 200 Rayleighs, which is comparable to typical red-line airglow emissions. Even on a moonless night, continuum emissions can be on the order of 10 R/nm, equivalent to stars of magnitude 2 as observed by our MSP. Note that there are only 50 stars of magnitude 2 or brighter, and fewer than half of them are visible from the northern auroral zone at any given time. ""

Page 9 table 2: [sm2 nm]??1 is centered above [J] and [#] looks strange.

—True. Moved units into table caption.

page 10 lines 14ÂŰ15: "...is still a hundred times brighter than the brightest aurora." IBC-IV aurora (1 MR at 557.7 nm) is often compared to the luminous intensity of the full moon (0.1 Lux for a human observer) . This doesnŠt make sense with "a hundred times brighter"

—Good point. We mistakenly used 100kR as an upper limit for auroral brightness. New text: " Despite this substantial decrease, the equivalent lunar brightness of nearly 10 megaRayleighs per nanometer (Table 2) is still 10 times greater than the brightest aurora (1 megaRayleigh for IBC-IV). " After some cross-checking of Table 2 (below), I'm reasonably confident that the energy and number flux values are correct.

1) The solar energy flux at 556 is 1.81 J/s/m^2/nm from Table 2, which is consistent with Figure 3. 2) The ratio of moon to sunlight from Table 2 is about 391000, which is consistent with the widely quoted astronomical difference in magnitudes of 14 (2.51^14 $\sim$ 398000). 3) Assuming the energy of a green photon to be 4e-19 Joules gives the same number flux as Table 2.

If so, then neglecting atmospheric absorption the full moon differential irradiance will be roughly 1300 x 1e10 photons/m^2/second/nm at 556nm. If those photons were isotropically distributed then the radiance would be L = 100 x 1e10 photons/mˆ2/second/nm per steradian or 1300 Rayleighs per nanometer.

For an MSP with 0.002 steradian FOV the result is roughly 8 MR/nm. If the moon completely fills the field of (6.67e-5 steradians) then then equivalent brightness is 250 MR/nm.

Looking at Chaimberlain Appendix II the IBC-IV is defined just in terms of 1e6 Rayleighs. Later sources (eg. Hargreaves) add qualitative descriptions such as "Full Moonlight". However, it is not clear what FOV or bandwidth are intended. Without this information it is difficult to make a quantitative comparision. Further comments regarding this (or any other point) would be appreciated.

page 17 Eq. (30): No reason to use the inverse of Eq(29).

—Agreed.

page 28, Eq(34): This equation is central to the paper and should be discussed in greater detail.

—Most of the next page is spent discussing the terms which contribute to this equation. Is more detail required? Is something missing?

page 29 line 12 ÂŰ page 31 line 7: Please consider moving (parts of) this text to the introduction.

—Yes. Paragraph with key references is now in the introduction where it belongs.

page 32 lines 6ÂŰ8 "An arc moving from the horizon to zenith will become brighter, not because of any change in precipitation, but simply due to reduction in total airmass between auroral altitudes and a ground-based observer." Correct, but please also consider number of photons integrated when looking along the magnetic field-line instead of across it. This is the main cause of the intensification.

—Agreed. New text: " A constant emission feature moving from the horizon to zenith

will appear brighter even after accounting for viewing geometry (i.e. Van Rhijn correction) simply due to the reduction in total airmass between auroral altitudes and a ground-based observer. "

page 33 conclusions: Maybe summarize a bit better, and/or include a small table of the most important results. Future outlook?

—Yes, conclusions expanded and (hopefully) improved.

Table 8: clarify units.

—Done.

Figure 1: reproduces badly and lacks site mnemonics (RANK, GILL, etc.)

—Figure re-done.

Figure 2: Keogram empty in printout. Looks good in PDF.

—Figure re-done.

Figure 8. x and y labels in the figure could be improved

—Done.

Please also note the supplement to this comment:
http://www.geosci-instrum-method-data-syst-discuss.net/gi-2016-5/gi-2016-5-AC2-supplement.pdf

**Supplement:**

**Auroral meridian scanning photometer calibration using Jupiter**

Brian J. Jackel[1], Craig Unick[1], Fokke Creutzberg[2], Greg Baker[1], Eric Davis[1], Eric F. Donovan[1], Martin Connors[3], Cody Wilson[1], Jarrett Little[1], M. Greffen[1], and Neil McGuffin[1]

[1]University of Calgary, Alberta, Canada
[2]Natural Resources Canada Geomagnetism Laboratory
[3]Athabasca University, Alberta, Canada

*Correspondence to:* Brian J. Jackel
brian.jackel@ucalgary.ca

**Abstract.** Observations of astronomical sources  provide information that can significantly enhance the utility of auroral data for scientific studies.  This report presents results obtained by using Jupiter for field cross-calibration of 4 multi-spectral auroral meridian scanning photometers during 2011-15 northern hemisphere winters. Seasonal average optical field-of-view and local orientation estimates are obtained with uncertainties of $0.01°$ and $0.1°$ respectively. Estimates of absolute  sensitivity are repeatable to roughly 5% from one month to the next, while the relative response between different wavelength channels is stable to better than 1%. Astronomical field calibrations and darkroom calibration differences are on the order of 10%. Atmospheric variability is the primary source of uncertainty; this may be reduced with complementary data from co-located instruments.

**1 Introduction**

Interactions between the solar wind and the terrestrial magnetic field produce a complex and dynamic geospace environment. Ionospheric phenomena such as the aurora are connected to magnetospheric processes by mass and energy transport along magnetic field lines. Consequently, auroral observations provide information that can be used for remote sensing of distant plasma structure and dynamics. A single ground-based instrument can only view a small part of the global system. , so a combination of instruments at different locations (eg. Figure 1 and Table 1)  multiple data sets requires accurate information about device characteristics such as timing, orientation, and absolute spectral sensitivity.

Comprehensive calibration requires specialized equipment and skilled personnel that are typically available only at centrally located research facilities. With sufficient resources it is possible, at least in principle, to determine all device parameters that are required to convert raw instrument data numbers to physically useful quantities. Practical limitations can result in random or systematic uncertainties which may impede quantitative scientific analysis. This is particularly relevant for large networks of nominally identical instruments, where ongoing calibration of each device may be extremely challenging.

Even assuming ideal calibration at a central facility, many auroral instruments must be operated at remote field sites. Transfer between these locations requires a sequence of packing, shipping, and re-assembly that is time-consuming, costly, and may

[Figure]

**Figure 1.** Canadian meridian scanning photometer site locations (details in Table 1). Fan shapes indicate $4°$ optical beam width for altitudes of 110 and 220 km at elevations of $10°$ above the horizon.  Dashed contours indicate magnetic dipole latitude (IGRF 2015).

**Table 1.** Canadian meridian scanning photometer site information. Geographic latitude, longitude, and altitude are in degrees North, degrees East, and metres above mean sea level (WGS-84). L-shell and magnetic declination  from the IGRF model.

| | Geographic | | | L-shell | | Declination | | |
|---|---|---|---|---|---|---|---|---|
| | Lat | Lon | Alt | 1988 | 2013 | 1988 | 2013 | |
| RANK | 62.82 | 267.89 | 32 | 11.20 | 10.64 | -7.1 | -7.7 | Rankin Inlet, Nunavut |
| GILL | 56.35 | 265.29 | 99 | 6.04 | 5.83 | 2.6 | -0.5 | Gillam, Manitoba |
| PINA | 50.20 | 263.96 | 262 | 3.95 | 3.84 | 5.5 | 2.3 | Pinawa, Manitoba |
| FSMI | 60.02 | 248.05 | 205 | 6.65 | 6.58 | 24.3 | 15.8 | Fort Smith, NWT |
| ATHA | 54.70 | 246.70 | 533 | 4.50 | 4.45 | 21.1 | 15.3 | Athabasca, Alberta |

unintentionally alter instrument response. Furthermore, intermittent calibration cannot distinguish between a gradual drift or sudden changes.

Extra-terrestrial sources, such as planets or stars,  are often used for calibration of spatially resolved optical or radio frequency data. Instrument orientation can be determined from objects whose positions are well known, while source inten-
5  sity can be used to verify instrument sensitivity. Astronomical sources are often detectable in existing auroral data streams, allowing for ongoing monitoring of system response and the possibility of retrospective re-analysis of older data sets. Practical application may be restricted by instrumental limitations and complications including man-made interference, clouds, aurora and other geophysical processes.

There is a long history of using astronomical sources to determine the alignment of auroral instruments (**????**). Absolute
10  calibration using stellar spectra appears to be a more recent development (**??????**). Detailed discussions of these topics are not always provided in the primary scientific literature, but must often be extracted from conference proceedings, technical reports, and theses.

The focus of this paper is on the field calibration of a network of four auroral photometers using Jupiter as a standard reference. Some key features of optical aurorae are provided in Section 1.1, §1.2 introduces key calibration concepts and
15  results, essential astronomical topics are presented in §1.3, and atmospheric effects are briefly reviewed in §1.4. An overview of instrument details is given in §2, data analysis and results are in §3, discussion in §4, followed by a summary and conclusions in §5.

**1.1 Optical Aurora**

In regions of geospace where magnetic field lines can be traced to the Earth, some charged particles may travel down to
20  altitudes where neutral densities are no longer negligible. Collisions with atmospheric atoms or molecules may transfer energy which can be re-emitted as photons. Spectral, spatial, and temporal features of the optical aurora contain information about geospace plasma properties, allowing for remote sensing of magnetospheric topology and dynamics.

Auroral spectra are dominated by several  relatively bright lines and bands from atomic oxygen and molecular nitrogen, with many other less intense features ranging from extreme ultra-violet through to far infra-red. The intensity of auroral emission at different wavelengths depends on precipitation energy and atmospheric composition, as more energetic particles are able to penetrate to lower altitudes where constituents may be more or less abundant. Consequently, observations at multiple wavelengths can be combined to infer characteristics of the precipitating particles (**??**). These multi-spectral measurements can be challenging due to the wide dynamic range between very bright 558 nm "green-line" (1-100 kiloRayleigh) emissions and extremely faint 486 nm "proton aurora" ($<$ 100 Rayleighs).

Optical aurora typically occur within "auroral ovals", roughly centered around each geomagnetic pole, extending hundreds of kilometres in latitude and thousands of kilometres in longitude (**?**). Luminosity can be highly dynamic over a wide range of spatial scales, but quiet-time structures generally exhibit a narrow latitudinal extent (10's to 100's of km) and relatively less longitudinal variation over 100's or 1000's of km (**?**). This spatial anisotropy is one motivation for using a meridian scanning photometer (MSP, see §2) to measure auroral luminosity as a sequence of latitude profiles (keogram). As shown in Figure 2, this data can also be used to identify other non-auroral features such as clouds and stars.

**1.2 Instrument Calibration**

Optical designs can be  modeled very precisely with modern software tools, but instrument calibration provides essential information about the actual performance. System response is not necessarily constant, but can change either gradually (eg. filter bandpass drift, decreased detector sensitivity) or abruptly (eg. damage during shipping). Such problems could be identified with calibration of instruments in the field. This process must be completely automatic, as many remote sites do not have full-time technical staff. It should be repeated frequently in order to identify abrupt changes in system response, but without interrupting or degrading normal data acquisition. A regular schedule of measurements with portable low-brightness sources (LBS) might satisfy some of these requirements, but would involve a substantial allocation of resources for repeated site visits.

In this report we examine some of the strengths and limitations of astronomical calibration for auroral instruments. We focus on issues related to field cross-calibration of MSPs which have been used extensively for auroral research (see §2 for details). However, many of these topics can also be applied more generally to other instruments used to study the optical aurora, such as all-sky imagers (ASIs).

A single ground-based instrument may measure photons with wavelengths $\lambda$ arriving from angular locations $\theta, \phi$.  The distribution of incident light $I$ is convolved with the instrument response function $f$ to product a measurement $M$ with error $M_\epsilon$

$$M(\theta, \phi, \lambda) = f(\theta, \phi, \lambda) \star I(\theta', \phi', \lambda') + M_\epsilon(\theta, \phi, \lambda) \tag{1}$$

For an ideal device $f$ would be a delta function and $M = I$, but any real measurement will have limited resolution. The goal of calibration or characterization is to determine the instrument response function $f$ in order to better understand the "true" source properties.

[Figure]

**Figure 2.** Keogram from meridian scanning photometer operating at Gillam during the night of December  20 2012 from  0000 UT  to dawn at 1320. Local midnight is approximately 0600 (scan number 720).  Data counts have been clipped and  log-scaled in  to  display Jupiter, stars, aurora, full moon, and  dawn.

[revised manuscript text omitted]

**1.2.3 Radiometry**

 At a distance $R$ from an isotropic point source with total power output  $P_0$  the irradiance (intensity) $S$  will be

$$S = \frac{P_0}{4\pi R^2} \qquad \text{watt} \cdot \text{meter}^{-2} \tag{8}$$

so that an observer at some distance $r$ will intercept an amount of power

$$P_\delta = S\, A_{\text{eff}} \tag{9}$$

proportional to the effective receiver surface area $A_{\text{eff}}$ .

Power from an extended source can be expressed in terms of a volume emission rate $\rho(r,\theta,\phi)$ integrated over the entire source region weighted by the receiver angular sensitivity $G(\theta,\phi)$

$$P_V = \oiint d\Omega\, \frac{L\,G}{4\pi} \qquad 4\pi L \equiv \int_0^\infty dr\, \rho(r) \tag{10}$$

where the radial integral $L$ has units of radiance (watt $\cdot$ meter$^{-2}$ $\cdot$ sr$^{-1}$) and is often referred to as the "column emission rate". For a uniform source radiance the total received power

$$P_V = \oiint d\Omega\, \frac{L(\theta,\phi)}{4\pi} A_{\text{eff}} \hat{G}(\theta,\phi) \approx L\ A_{\text{eff}}\, \Omega_0 \tag{11}$$

depends on the product of the effective area and the effective solid angle.  For any signal detected from some point source  there will be an equivalent volume emission which would produce the same observed power. For a uniform emission region the  relationship

$$P_\delta = P_V \qquad \rightarrow \qquad L = \frac{S}{\Omega_0} \tag{12}$$

depends only on the effective solid angle.

Auroral  intensity $\mathcal{I}$ is customarily expressed in units of Rayleighs (**????**) which is related to photon radiance $L_\gamma$ via

$$4\pi L_\gamma(\lambda) \equiv \mathcal{I}(\lambda) \qquad 10^{10}\ \text{photon} \cdot \text{s}^{-1} \cdot \text{m}^{-2} \tag{13}$$

where the subscript $E$ indicates energy flux and $\gamma$ is photon number flux. For narrow-band channels

$$\mathcal{I}(\lambda) = \int \dot{\mathcal{I}}(\lambda) \approx \dot{\mathcal{I}}\, \Delta\lambda = 4\pi \frac{\dot{S}_E}{\Omega_0} \frac{\lambda}{h\,c} \Delta\lambda \tag{14}$$

converting differential radiant spectral density $\dot{S}$ to equivalent Rayleighs per nanometer $\dot{\mathcal{I}}$ requires only the effective solid angle, which can also be estimated from observations of a point source. Working with Rayleighs requires some additional knowledge in the form of the effective bandwidth $\Delta\lambda$. As this is also true for darkroom LBS calibration, we focus here on relating $\dot{I}$ in Rayleighs per nanometer to $\dot{S}$ in Watts per metre-squared per nanometer.

**1.3 Astronomical sources**

Extra terrestrial objects have many properties which are required for accurate calibration. Locations in the celestial sphere are known to arc-second resolution or better, which is  sufficient for determining orientation and geometric response of most auroral instruments. Absolute spectral irradiance profiles are available for many sources, providing opportunities for  radiometric calibration of narrow-band instruments. Total visible intensity of most sources is essentially constant, allowing for long term monitoring of system performance. A single object can be viewed simultaneously by multiple instruments at nearby sites, facilitating quantitative inter-comparisons.

Most astronomical objects are effectively point sources,  and under good viewing conditions modern all-sky imagers can resolve hundreds of stars with a relatively short exposure time. Ironically, the presence of bright aurora or airglow can be a major source of error in radiometric calibration. For the MSP considered here, the total light from Vega passing through through a 3 nm filter is approximately 200 Rayleighs, which is comparable to typical red-line airglow emissions. Even on a moonless night, continuum emissions can be on the order of 10 R/nm, equivalent to stars of magnitude 2 as observed by our MSP. Note that there are only 50 stars of magnitude 2 or brighter, and fewer than half of them are visible from the northern auroral zone at any given time.

Celestial source brightness spans a wide range and is usually expressed in terms of logarithmic magnitude $m$

$$I = \sqrt[5]{100}^m \approx 2.512^m \tag{15}$$

so that the relative intensity of two sources can be determined from the difference of their magnitudes. Absolute flux distributions as a function of wavelength are available for most of the brightest stars, including Vega (**?**), Sirius (**?**), and Arcturus (**??**). Other catalogs contain many other stars (**?????**), but the majority may be too dim for reliable observation by typical auroral instruments.

**Table 2.** Selected astronomical source irradiance at Earth. Energy flux is Joules per $[s \cdot m^2 \cdot nm]$ and number flux is photons per $[s \cdot m^2 \cdot nm]$ Rayleighs are for a viewing solid angle of $\Omega = 0.002$ steradians ($2.9°$ of arc).

|         | [nm] | [$J$]    | [#]      | [R / nm] |
|---------|------|----------|----------|----------|
| jupiter | 486  | 4.78e-10 | 1.17e+09 | 735      |
| jupiter | 556  | 5.45e-10 | 1.53e+09 | 958      |
| sirius  | 556  | 1.35e-10 | 3.78e+08 | 237      |
| vega    | 556  | 3.44e-11 | 9.63e+07 | 60.5     |
| moon    | 556  | 4.63e-06 | 1.3e+13  | 8.14e+06 |
| sun     | 556  | 1.81     | 5.07e+18 | 3.18e+12 |

Conversely, the sun is so bright that direct observation will saturate detectors designed for relatively faint aurora. **?** provide an absolutely calibrated distribution of flux versus wavelength at 1 AU with sub-nanometer spectral resolution. For a nominal

instrument solid angle of 2 milli-steradians (3° of arc) the apparent solar brightness at 556 nm is roughly 3 teraRayleighs per nanometer (Table 2). Daytime operations are only possible for systems that respond to an extremely narrow range of wavelengths (**?**).

[Figure]

**Figure 3.** Spectra of solar irradiance ( green shaded curve)  from **?** and Jupiter albedo (blue line) from **?**. Inset  displays the same quantities  for the range of wavelengths associated with most visible aurora.

[revised manuscript text omitted]

detection of narrow emission lines.  An effective passband

$$\Delta\lambda_j = \int d\lambda\, \hat{T}_j(\lambda) \tag{25}$$

is the relevant quantity for broad-band calibration sources ie. converting from Rayleighs per nanometer to Rayleighs. These data suggest typical passband and transmission variations on the order of 5% between different sets of filters.

**Table 5.** Characteristics of three sets of nominally identical narrow band filters. Passband is integral of transmission profile, 90% bandwidth is the range between 5% and 95% points of the cumulative transmission.

| [nm] | passband [nm] | | | peak transmission [%] | | | 90% bandwidth [nm] | | |
|---|---|---|---|---|---|---|---|---|---|
| 470.9 | 2.483 | 2.362 | 2.355 | 82.94 | 79.36 | 78.28 | 3.60 | 3.40 | 3.50 |
| 480 | 2.592 | 2.418 | 2.661 | 85.59 | 78.60 | 87.74 | 3.10 | 3.20 | 3.20 |
| 486.1 | 2.605 | 2.587 | 2.615 | 88.26 | 85.95 | 87.57 | 2.90 | 3.00 | 3.00 |
| 486.1 | 2.572 | 2.222 | 2.509 | 84.71 | 74.21 | 83.44 | 3.10 | 3.00 | 3.10 |
| 495 | 2.607 | 2.525 | 2.584 | 88.40 | 85.74 | 87.27 | 3.30 | 3.40 | 3.50 |
| 557.7 | 1.788 | 1.728 | 1.920 | 82.93 | 78.93 | 88.53 | 4.60 | 4.90 | 4.30 |
| 625 | 1.624 | 1.632 | 1.588 | 84.46 | 87.37 | 86.09 | 4.20 | 3.20 | 2.80 |
| 630 | 1.597 | 1.590 | 1.558 | 86.67 | 84.83 | 83.81 | 2.40 | 2.60 | 2.40 |

Light which passes through the filters is detected by a photo-multiplier tube (PMT) with photocathode quantum efficiency ranging from 20% at 400 nm to 2% at 750 nm; this response was selected to maximize response for the faint $H_\beta$ emissions. A dynode chain amplifies each electron to produce a cascade which triggers a pulse-counting circuit. The high-voltage power supply required for this process is quite stable over short intervals under ideal conditions, but may change during extended field operations. Photocathode aging and high-voltage drift are likely to be the primary  causes of any long-term reduction in system sensitivity.

PMTs dead-time produces a non-linear response at high count-rates. This pulse pile-up effect can be largely removed if the time resolution $\tau$ of the system is known and is not significantly longer than the signal count interval. For the PMTs used in this study nonlinearity only becomes important for count rates greater than $10^5$ photons per second. These rates can be produced by very bright aurora but are not a problem for any astronomical sources except the Sun and Moon.

Meridian scans are achieved with a $45°$ tilted mirror and a stepping motor. Many MSPs rotate the mirror at a fixed rate in order to produce data from evenly spaced elevations. Both the original and refurbished systems considered here instead utilize a sequence of variable steps chosen to produce nearly constant exposure times as a function of linear distance at auroral altitudes. This detail is relevant to this study because Jupiter transit profiles will be measured with different resolution depending on transit elevation. The effects are expected to be small, but must be kept in mind when considering multi-year variability.

**2.1 System sensitivity**

The relationship between incident photon flux $\mathcal{P}(\lambda)$ and measured channel count rate $\mathcal{D}_k$

$$\mathcal{D} = A_{\text{eff}}\, M_x\, \Delta t \int d\lambda\, \mathcal{P}(\lambda)\, T_k(\lambda)\, Q(\lambda) \tag{26}$$

depends on the effective aperture allowing photons into the system ($A_{\text{eff}}$), channel multiplexing efficiency ($M_k$), filter transmission ($T_k$), measurement interval ($\Delta t$), and the detector efficiency $Q(\lambda)$.

For wide-band input through narrow-band filters the process can be written in terms of filter peak transmission $T_k$ and bandwidth $\Delta\lambda_k$

$$\mathcal{D}(\lambda_i) \approx \mathcal{P}(\lambda_k)\, A_{\text{eff}}\, M_k\, \Delta\lambda_k\, T(\lambda_k)\, \Delta t\, Q(\lambda_i) \tag{27}$$

 from which we can isolate a coefficient of response $_k\mathcal{C}_{\mathcal{D}/\mathcal{P}}$ for each channel

$$_k\mathcal{C}_{\mathcal{D}/\mathcal{P}} = \frac{\mathcal{D}(\lambda_k)}{\mathcal{P}(\lambda_k)}$$
$$= A_{\text{eff}}\, M_x\, \Delta\lambda_k\, T(\lambda_k)\, \Delta t\, Q(\lambda_k) \tag{28}$$

in terms of measured $\mathcal{D}$ and predicted $\mathcal{P}$ for each filter wavelength.  In principle this equation could be used to calculate coefficients in terms of  fundamental properties of each instrument. In practice, calibration coefficients are often estimated empirically by measuring sources with known brightness. For auroral applications the goal is to determine the differential sensitivity $\mathcal{C}_{\mathcal{D}/\mathcal{R}}$ relating data numbers to Rayleighs per nanometer.

[revised manuscript text omitted]

Fortunately, it is possible to accurately determine instrument orientation from transit observations. Starting with site locations obtained using GPS, observed transit times were used to calculate the actual elevation and azimuth of Jupiter for each night. These were interpreted in terms of two device angles. First, azimuth offset was attributed to horizontal orientation of a level instrument. Second, the difference between nominal mirror elevation and actual target elevation was attributed to instrument 5 "tilt" from level.

Results for in Figure 9.  several seasons of azimuth estimates at Gillam (not shown) are extremely stable over time, with jitter $< 1°$ and no apparent drift. Tilt estimates  are shown in in Figure 9. The first two seasons are generally stable, although there appears to be a small jump in early November. Examination of results from the other three sites (not shown) finds a similar feature at Fort Smith (FSMI), a smaller shift 10 at Pinawa (PINA), and no obvious change at Athabasca (ATHA). These results are consistent with "frost heave" occurring in early winter as moisture in the soil freezes. The lack of this effect at ATHA may be be due to better foundations for the instrument platform. The large change in tilt at Gillam during summer 2013 occurred around the same time as a maintenance trip. This shift could not have been detected in real-time due to the lack of Jupiter transit data during limited observing hours during summertime operations. Fortunately, once the problem has been identified, it is relatively straightforward to make the 15 necessary corrections to scientific data products.

[revised manuscript text omitted]
  extended luminosity. This can be related to the differential irradiance of an ideal point source using Equation 14. Losses due to atmospheric effects can be

[Figure]

**Figure 12.** Ratio of 630.0 nm to average of two 486.1 nm channels versus time. Large symbols correspond to good transits and small symbols to noisier events.

modelled with Equation 20. The combination of these three equations

$$\mathcal{C}_{\mathcal{D}/\dot{\mathcal{R}}} = 10^{10} \frac{\mathcal{D}}{\dot{S}_\gamma} \frac{\Omega_0}{4\pi} 2.512^{+\kappa X} \tag{32}$$

 gives an expression for calibration coefficients in terms of five physical quantities (see also page 42 of **?**). Three of these terms are easily estimated, while the other two present some challenges.

5    The differential number flux $\dot{S}_\gamma$ of solar photons scattered from Jupiter and arriving at the top of the Earth's atmosphere is only subject to uncertainties in the solar spectrum and Jupiter's albedo, both of which are known to 1% or better. The effective air-mass $X(\zeta(t))$ depends on the apparent zenith angle which can be calculated for any arbitrary time. The effective solid angle $\Omega$ is either known *a priori* or can be estimated from transit profiles.  ; from §3.1 the uncertainty of an unbiased estimate will be less than 1%, but systematic bias on the order of 5% is also a possibility.

**Table 8.** Magnitude normalized intensity and self-normalized spectral sensitivity for Gillam and Fort Smith. Column 4 is the 90th percentile of intensity. Column 4 is the source normalized brightness (Equation 30). Remaining columns are channel brightness normalized to average of two 486 nm observations.

[revised manuscript text omitted]
. A constant emission feature moving from the horizon to zenith will appear brighter even after accounting for viewing geometry (i.e. Van Rhijn correction) simply due to the reduction in total airmass between auroral altitudes and a ground-based observer. Atmospheric effects may be negligible when looking directly upward through clear skies, but critically important at low elevations and non-ideal viewing conditions. These effects would be even more pronounced at shorter wavelengths (eg. 427.8 nm and 391.4 nm) often used in auroral studies.

**4.2 Retrospective Calibration**

Some auroral instruments only acquire data during short-term "campaigns", but many are operated in support of longer term science objectives. Not all devices are fully calibrated before being deployed and few are calibrated on a regular basis. Even when the resulting data overlap in space and time, quantitative comparison may not be possible. Astronomical observations of bright sources such as Jupiter can provide a basis for retrospective cross-calibration of historical data sets.

The original CANOPUS meridian scanning photometer array (MPA) is a good example. Digital "low resolution" binned data are available starting in early 1988 and continuing until spring 2005. Some higher resolution data are available for the transition period from 2005-2010, after which all refurbished instruments were operating in the same high-resolution mode. The 16 years of low-res data alone extend well beyond one solar cycle and could span more than two if merged with newer data.

However, certain kinds of quantitative analysis are limited by the lack of radiometric calibration. Some key parameters (eg. filter band-width and channel sensitivity) were determined for each system, but the supporting documentation is very limited. Mechanical and electrical subsystems were regularly maintained and repaired, but there was no corresponding re-calibration schedule. Some terminal calibration procedures were carried out during the 2005-2010 transition, but by this point the instruments were often not functioning reliably. In order to confidently identify long-term geophysical trends in these data it is essential to have some sense of how instrument performance changed over the same time-scales.

A preliminary survey of the CANOPUS MPA data archive has confirmed the feasibility of astronomical calibration and also identified some significant challenges. First, only the brightest few stars are visible even with optimal viewing conditions. Jupiter can be clearly identified, but at count rates much lower than obtained by the newer systems, and consequently with much greater uncertainty. Elevation steps are combined into 17 latitude bins which effectively removes the ability to determine instrument tilt. More generally, it eliminates virtually all information about the optical beam-shape in that direction, including that required to confidently estimate the effective solid angle $\Omega_0$. Finally, the decreased scan cadence of one-per-minute will slightly reduce the accuracy of azimuth estimates. Despite these limitations it should still be possible to estimate absolute

sensitivity using Jupiter transits during extended intervals at both ends of the project: 1989-1993 and 1999-2005. Other bright stars or planets might be used to fill in the intervening period.

**5 Conclusions**

In this study we have demonstrated the feasibility of using Jupiter to calibrate a network of auroral meridian scanning photometers.  During times when Jupiter is visible in the night sky it can be easily distinguished from other astronomical sources. Statistical uncertainty may be a limiting factor even for bright stars, so the increased signal from Jupiter is highly advantageous. Addition precision can be achieved by combining results from multiple days with good viewing conditions.

For geometric calibration, this approach provides an estimate of instrument orientation for each transit with even marginal viewing conditions.  Changes of less than $1°$ between successive transits can be easily identified.  Absolute orientation can be determined to at least $1/10°$, which exceeds most application requirements. Angular optical response (beam-shape) can be  estimated to roughly 1% precision by combining several dozen transits.

Relative spectral calibrations (ratios of different channels) can also be obtained with precisions on the order of 1% during a single field season. Absolute radiometric calibration for individual channels is significantly less precise. This is due primarily to the difficulty of obtaining and identifying perfectly "clean" transits. Even results from apparently ideal transits can differ by 5-20%, likely due to uncertainties in the true atmospheric extinction parameters.

The merits of Jupiter as a calibration source also apply to other types of auroral instruments. Utility of stellar calibration for all-sky imagers has been demonstrated (**???**) and these methods would be even more effective with a brighter source. Given the complexities of absolute calibration, it might be helpful if observations were presented in some standard format, eg. data numbers normalized to source magnitude $\mathcal{D}_0$ as defined in Equation 30. This, along with estimates of solid angle $\Omega_0$ and bandwidth $\Delta\lambda$, would greatly facilitate the inter-comparison of different data products, which would be beneficial for both instrument operators and end-users of scientific data products.

In principle, astronomical calibration could be extracted from almost any auroral data set. In practice, this process is typically applied on a case-by-case basis and requires a considerable amount of human intervention and instrument specific knowledge. The essential next step is to develop automated software tools which can be applied more broadly. This will significantly increase the utility of optical auroral observations for quantitative scientific analysis.

*Acknowledgements.* Funding for  MSP refurbishment and ongoing operation was provided by the Canadian Space Agency under Go Canada initiative contract 13SUGOHSTO for the H STORM project. Field operations support is provided by SED systems and Athabasca University.